# Artificial Intelligence for Risk Stratification in Diffuse Large B-Cell Lymphoma: A Systematic Review of Classification Models and Predictive Performances

**DOI:** 10.3390/medsci13040280

**Published:** 2025-11-24

**Authors:** Dragoș-Claudiu Popescu, Mihnea-Alexandru Găman

**Affiliations:** 1Faculty of Medicine, “Carol Davila” University of Medicine and Pharmacy, 050474 Bucharest, Romania; 2Bone Marrow Transplantation Ward, University Emergency Hospital Bucharest, 050098 Bucharest, Romania; 3Department of Cellular and Molecular Pathology, Stefan S. Nicolau Institute of Virology, Romanian Academy, 030304 Bucharest, Romania; mihneagaman@yahoo.com; 4Department of Hematology, Center for Clinical and Basic Research (CCBR Clinic), 030463 Bucharest, Romania

**Keywords:** diffuse large B-cell lymphoma, DLBCL, machine learning, artificial intelligence, prognosis, risk stratification, deep learning, radiomics, transcriptomics, circulating tumor DNA, multi-omics integration, digital pathology, review

## Abstract

**Background**: Diffuse large B-cell lymphoma (DLBCL) is a biologically heterogeneous malignancy, with various outcomes despite significant advances in therapeutic options. Current conventional prognostic tools, e.g., the International Prognostic Index (IPI), lack sufficient precision at an individual patient level. However, artificial intelligence (AI), including machine learning (ML) and deep learning (DL), can enable specialists to navigate complex datasets, with the final aim of improving prognostic models for DLBCL. **Objectives**: This scoping review aims to systematically map the current literature regarding the use of AI/ML techniques in DLBCL outcome prediction and risk stratification. We categorized studies by data modality and computational approach to identify key trends, knowledge gaps, and opportunities for their translation into current practice. **Methods**: We conducted a structured search of the PubMed/MEDLINE, Scopus, and Cochrane Library databases through July 2025 using terms related to DLBCL, prognosis, and AI/ML. Eligible studies included original papers applying AI/ML to predict survival outcomes, classify risk groups, or identify prognostic subtypes. Studies were categorized based on input modality: clinical, positron emission tomography/computed tomography (PET/CT) imaging, histopathology, transcriptomics, genomics, circulating tumor DNA (ctDNA), and multi-omics data. Narrative synthesis was performed in line with PRISMA-ScR guidelines. **Results**: From the 215 records screened, 91 studies met the inclusion criteria. Group-wise we report the following categories: clinical risk features (n = 8), PET/CT imaging (n = 30), CT (n = 1), digital pathology (n = 3), conventional histopathology (n = 2), gene expression profiling (n = 19), specific mutational signatures (n = 18), ctDNA (n = 3), microRNA (n = 2), and multi-omics integration (n = 5). The most common techniques reported amongst the papers included ensemble learning, convolutional neural networks (CNNs), and LASSO-based Cox models. Several AI techniques demonstrated superior predictive performance over IPI, with area under the curve (AUC) values frequently exceeding 0.80. Multi-omics models and ctDNA-based predictors showed strong potential for clinical translation, a perspective worth considering in further studies. **Conclusions**: AI/ML methods are increasingly used in DLBCL to improve prognostic accuracy by leveraging data types with diverse inputs. These approaches allow an enhanced stratification, superior to traditional indices, and support the early identification of high-risk patients, earlier guidance for therapy tailoring, and early trial enrollment for flagged cases. Future investigations should focus on external validation and improvement of model interpretability, with tangible perspectives of integration into real-world workflows and translation from bench to bedside.

## 1. Introduction

Diffuse large B-cell lymphoma (DLBCL) is the most common subtype of non-Hodgkin’s lymphoma, characterized by clinically aggressive behavior and a heterogeneous biological landscape [1]. The current standard of care involves a chemoimmunotherapy backbone regimen, either R-CHOP (rituximab, cyclophosphamide, doxorubicin, vincristine, and prednisone) or pola-R-CHP (CD79b-directed antibody drug conjugate polatuzumab vedotin-piiq plus R-CHP), which has a near 60% curability rate. Nevertheless, a significant proportion of subjects will most likely fall into the category of primary refractory DLBCL or will relapse in the first 24 months after therapy discontinuation [1]. Undoubtedly, these patients face a poor prognosis, with lower survival rates for those who do not undergo a curative salvage therapy, such as stem cell transplantation or chimeric antigen receptor T-cell (CAR-T) therapy. According to the most recent version of the National Comprehensive Cancer Network (NCCN) B-cell lymphoma treatment guidelines, DLBCL patients who are primary refractory or who relapse less than 12 months after therapy discontinuation can be referred to CD19-directed CAR-T therapies (axicabtagene ciloleucel, lisocabtagene maraleucel) and receive bridging therapy as required until the CAR-T product is available. Those who are not CAR-T cell therapy candidates can be treated with CD20-directed bispecific antibodies (epcoritamab, glofitamab), conventional chemotherapy, or other options. DLBCL patients who experience disease relapse after 12 months from therapy discontinuation and are strong candidates for stem cell transplantation should receive aggressive chemotherapy as a bridge to transplant, whereas those who are not strong candidates for the procedure can benefit from CD19-directed CAR-T cell therapy, bispecific antibodies, or other treatment regimens. CAR-T cell therapy and bispecific antibodies can also be used as therapeutic options in the third and subsequent treatment lines in DLBCL [2].

Since reliable prognostic tools for DLBCL are insufficient, different instruments based either on clinically oriented parameters (e.g., IPI, NCCN-IPI), molecular classification (e.g., cell-of-origin) and even more advanced genetic markers (e.g., MYC/BCL2 rearrangements, DHITsig, MCD/EZB clusters), have been tested [1]. However, individualization of DLBCL treatment plans according to such instruments remains limited as there are barriers in a comprehensive evaluation of disease markers in the clinic. Moreover, even when there are no impeding factors, there is still inter-patient molecular diversity and a lack of consensus on how to translate these findings from bench to bedside [3]. For example, recent investigations have shown that ctDNA could emerge as a promising biomarker in detecting early relapses and in DLBCL monitoring; however, it must be taken into account that its assessment requires notable financial costs and faces challenges in standardization [1].

As healthcare specialists have been searching for better diagnostic and prognostic tools to manage hematological malignancies, artificial intelligence (AI), particularly through machine learning (ML) techniques, has displayed the potential to fill in some unmet gaps. The major advantage of these particular algorithms lies in their ability to summarize and classify high-dimensional data from various sources—clinical records, histopathology, radiology, genomics, and liquid biopsies—augmenting the use-case of decisional trees and multiple layered networks—to obtain clinically relevant tools for prognosis and risk prediction [4,5]. In oncology and hematology, ML algorithms have proven to be potentially useful in several frameworks aimed at prediction of risk and categorization into subtypes with the final aim of implementing a more precise treatment protocol [6]. In particular, for DLBCL, AI/ML applications encompass various fields, from image-centered approaches (PET/CT techniques, digital histopathology) and molecular analysis (transcriptomics, mutational profiling) to integrative approaches such as multi-omics. However, current data is fragmented and only available in the setting of individual experiments.

This scoping review aims to extensively map the existing literature on the use of AI and ML for prognostic modeling in DLBCL. Our objective was to categorize studies by data modality and methodological approach, following key trends and performance outcomes, and identifying gaps which could lead to further iterations. Unlike traditional systematic reviews focused on effect estimates, our work prioritizes an investigational approach to classification, in line with scoping review methodology and PRISMA-ScR guidance. Encompassing advances across radiomics, digital pathology, omics, and risk modeling algorithms, we outlined a structured overview of how ML/AI was leveraged in various scenarios to address unmet needs in DLBCL.

Figure 1 illustrates the systematic categorization of 91 studies applying AI and ML for prognostic modeling in DLBCL.

## 2. Methods

This scoping review was conducted in accordance with the Preferred Reporting Items for Systematic Reviews and Meta-Analyses extension for Scoping Reviews (PRISMA-ScR) guidelines. The aim was to map and synthesize the existing literature on the application of AI, including ML and deep learning (DL), for prognostic modeling in DLBCL.

### 2.1. Eligibility Criteria

We included original research articles that applied AI, ML, or DL techniques to predict clinical outcomes, stratify risk, or enhance prognostic modeling in patients with DLBCL. Studies using imaging, clinical, pathological, transcriptomic, genomic, epigenomic, or circulating biomarkers as model inputs were eligible. In terms of article types, reviews, editorials, abstracts, conference posters, and studies that did not report performance metrics or relevant endpoints, e.g., overall survival (OS), progression-free survival (PFS), complete response, or molecular subtypes, were excluded.

### 2.2. Information Sources and Search Strategy

Our study involved a search in three major relevant electronic databases: PubMed/MEDLINE, Scopus, and Cochrane Library. The search strategy combined terms related to DLBCL, prognosis, and AI or ML concepts. The subsequent search strings, designed for comprehensive discovery, were applied:PubMed/MEDLINE: ((“Lymphoma, Large B-Cell, Diffuse”[Mesh]) AND “Prognosis”[Mesh]) AND “Artificial Intelligence”[Mesh]Scopus: TITLE-ABS-KEY ((“diffuse large B-cell lymphoma” OR DLBCL) AND (prognosis OR prognostic) AND (“artificial intelligence” OR “machine learning”))Cochrane Library: (“diffuse large B-cell lymphoma” AND prognosis AND “artificial intelligence”)

This review was conducted in accordance with PRISMA-ScR guidelines and was prospectively registered in PROSPERO (CRD420251083702). The final search was performed in July 2025.

## 3. Results

*Literature search*: A total of 215 records were identified across databases: 56 from PubMed/MEDLINE, 158 from Scopus, and 1 from the Cochrane Library. After deduplication, 180 unique articles were retained for screening. After screening of the 180 articles, 75 were excluded based on relevance, lack of outcome data, or absence of performance metrics. Discrepancies were resolved by consensus between investigators on a case-by-case basis. Of the remaining papers, 13 articles could not be retrieved in full and were excluded, and one article was excluded during full-text review. A final total of 91 articles were included in this review (Figure 2).

*Overview of Included Studies*: The 91 included studies were published between 2003 and 2025, reflecting a steady rise in the application of AI/ML in DLBCL prognostication. Studies were sorted and reported according to data manipulation method used or domain of application, as follows:**Clinical Features and Risk Scores** (n = 8)—Most of these studies reported logistic regression, ensemble models (e.g., XGBoost, LightGBM), or Bayesian networks to enhance prediction of OS and PFS based on data involving clinical and demographic features.**Digital Pathology and Optical Imaging** (n = 3)—Based on CNNs and MIL models that used hematoxylin and eosin (H&E)-stained whole slide images for diagnosis, decoded cell-of-origin (COO) classification, or served tools for therapy response prediction.**Conventional Histopathology** (n = 2)—Amongst these studies, ML algorithms were used on gene expression signatures or immunohistochemical surrogates to classify COO and predict survival.**CT conventional imaging** (n = 1)—A single study reports a model which focuses on secondary endpoints such as major cardiac events, estimated based on a calculated calcium score.**PET/CT Imaging** (n = 30)—A numerous cohort of studies using machine learning techniques such as deep learning and radiomics-based models to select various prognostic features derived from PET/CT scans. The studies mostly benchmarked their OS and PFS predictive scores to IPI models, outperforming them.**Transcriptomic and Gene Expression Profiling** (n = 19)—Many models included LASSO, autoencoders, and swarm optimization. The outputs ranged from immune subtype classification to gene signature-based survival prediction.**Specific Genetic Mutations** (n = 18)—Various supervised models, incorporating mutational data (e.g., TP53, MYD88, EZH2), aimed at risk stratification with refined molecular subtyping.**ctDNA and Liquid Biopsy** (n = 3)—A small number of reported studies employed ML to predict early relapse or survival using parameters such as the ctDNA burden, clonality, and even fragmentation features.**microRNA-Based Models** (n = 2)—Two studies which used miRNA expression to predict R-CHOP response via Random Forest classification and to diagnose a rare subtype of DLBCL via fluid expression of microRNA.**Multi-Omics Integration** (n = 5)—Integrative models which combine gene expression, methylation, CNVs, and clinical data using ML techniques (e.g., Random Forest, Cox regression) to produce advanced and refined composite prognostic scores.

### 3.1. Clinical Features and Risk Score Modeling

The current traditional prognostic tools in DLBCL include the IPI and NCCN-IPI scoring systems, which provide a generally accepted risk stratification but lack the ability to estimate the survival prognosis of an individual. Novel approaches could employ ML to enhance risk modeling from the routinely available clinical variables, similarly to traditional scoring systems, but with enhanced performances.

An investigation published by Biccler et al. aimed to assess the superiority of ML models over conventional scores in a cohort of over 10,000 lymphoma patients—of whom 4420 suffered from DLBCL [8]. The methods proposed by these researchers included ML models such as random survival forest and penalized Cox regression, which consistently outperformed the IPI score in predicting OS. It is notable to say that a simplified Cox model using only age and performance status managed to classify DLBCL patients better than the traditional IPI scale. Therefore, we can infer that there is a solid use case for non-dichotomized ML approaches for individualized prognostics in DLBCL [8].

In another study, also conducted by Biccler, the investigators used data from nationwide Danish and Swedish DLBCL cohorts totaling 5173 patients [8]. The ML methods used in this experiment combined Cox, AFT, and parametric models bundled in a stacked format, which outperformed IPI and NCCN-IPI in both cohorts, achieving a higher C-index (concordance index) (0.756 DK; 0.744 SE), and integrating lower Brier scores [8].

Fan et al. developed a prognostic score using logistic regression, which aimed to classify de novo DLBCL patients into high- and low-risk cases, using clinical and molecular features as inputs [9]. Moreover, the score served as an input for five other ML classifiers, i.e., Logistic Regression (LR), Random Forest (RF), Support Vector Machine (SVM), Naive Bayes (NB), and Feed-Forward Neural Network (FFNN), in order to further improve group stratification. Based on the datasets analyzed using their approach, the logistic regression model showed the best performance, scoring a C-index of 0.71 on external validation [9].

A rather different approach involving clinical data was conducted in a trial aimed at boosting the NCCN-IPI score itself. Kim et al. investigated whether the integration of stromal FOXC1 and tumor pERK1/2 expression with the NCCN-IPI score would benefit the prediction of actual survival. Therefore, they selected a cohort of 134 DLBCL patients treated with a standard R-CHOP regimen, for whom they applied six survival-specific machine learning algorithms (FastSVM, Random Survival Forest, XGBoost, and Bagging SurvTree) [10]. A 5-fold cross-validation and nine feature selection methods were applied. The time-to-death prediction results for the best performing model reported a C-index of 0.801 (*p* = 0.030), a result which, even tested on an internal cohort, highlights the importance of additional variables for prediction model enhancement [10].

Another assessment with a significant number of patients was based on information collected from the Surveillance, Epidemiology, and End Results (SEER) database and focused on pediatric DLBCL. The authors analyzed the data of 836 patients with DLBCL, applying an algorithm for prognostic model discovery through either Cox regression, generalized Cox regression, or XGBoost (Extreme Gradient Boosting) [11]. The prognostic features used as input data included: age, Ann Arbor stage, treatment modalities, and time from diagnosis to therapy. Qin et al. report that the XGBoost model achieved the best predictive performance, with an AUC of 0.892 (training settings) and 0.889 (validation), outperforming traditional methods. Therefore, we can state that this study demonstrated that in the pediatric population, prediction of overall survival in DLBCL could be improved by integrating ML techniques that leverage clinical data [11].

There were also other studies which looked at combining the IPI risk groups with immunohistochemical modifiers (namely the subtype—GCB and non-GCB) by using a classification and regression tree (CART) approach. In a study by Samarina et al., a cohort of 81 eligible patients treated with R-CHOP from Russian databases were stratified into low-, intermediate-, and high-risk groups. The results indicated prediction scores for the high-risk group at a 2-year OS with 46% accuracy, a median OS of 25 months, and for the low-risk group, achieved 100% accuracy in predicting 2-year and 5-year OS compared to the actual monitoring data available, as the authors report [12].

Besides the classical IPI scoring system, several researchers analyzed the medical records of DLBCL patients and investigated whether routinely documented parameters (demographic data, clinical stage, ECOG (Eastern Cooperative Oncology Group), GCB subtype, Hans algorithm, CBC (Complete Blood Count) indices, and serum markers such as albumin and cholesterol) could serve as independent predictors [13]. Based on data from 1211 newly diagnosed DLBCL cases from seven Chinese centers, Shen et al. retrospectively analyzed prognostic models for OS using both linear and non-linear ML methods. A penalized Cox regression model using LASSO (linear model) and a non-linear random forest model were compared, with LASSO achieving superior discrimination (AUC 75.8%, C-index 0.704) in the training cohort. In addition, they identified several independent predictors of OS in DLBCL, namely: age, WBC (White Blood Cell) count, hemoglobin, CNS (Central Nervous System) involvement, gender, and Ann Arbor stage [13].

Another innovative approach used data from the SEER registry and tackled a more challenging topic by selecting a rare subgroup of lymphomas, namely patients with composite Hodgkin’s lymphoma and DLBCL. Zhao et al. analyzed a subgroup of 869 patients, and applied six ML algorithms (XGBoost, Random Forest, AdaBoost, KNN, ANN, and GBDT). The prediction tests evaluated the 1-, 5-, and 10-year mortality, the best performing algorithm, returning an AUC > 0.80, being the XGBoost technique [14]. The authors also designed a Cox-based nomogram for OS prediction and validated all the results on external testing cohorts. While the XGBoost has proven superiority in these particular settings, the novel approach targeted a challenging group in clinical settings, i.e., composite lymphomas, and also provided tools which could be translated into clinical practice [14].

ML was also tested in the setting of relapsed/refractory DLBCL, aiming to build prognostic models for OS and PFS prediction using clinical and PET/CT data. The small study conducted by Zhu et al. aimed to investigate a subgroup of 28 Chinese DLBCL patients who would benefit from the R-ICE and ibrutinib combination. The study investigated four survival analysis methods (Random Survival Forest, Gradient Boosting Machine, Cox-XGBoost, and Stepwise Cox regression) [15]. The test cohort revealed time-dependent AUCs for OS prediction, which reached 0.863 (1 year), 0.864 (2 years), and 0.898 (3 years), while PFS AUCs ranged from 0.769 to 0.784. The paper also reports that the prognostic features that were most significant during the experiment were as follows: CD5+ expression, LDH (lactate dehydrogenase), ALB (albumin), β2-MG (beta-2 microglobulin), and time to relapse >12 months. Also, as seen in other studies, the best results for this type of application were obtained with a gradient boosting technique, Cox-XGBoost (AUC > 0.8) [15].

Table 1 summarizes the risk score models that were based on clinical features.

### 3.2. Digital Pathology and Optical Imaging

Deep learning (DL) models applied to whole-slide imaging (WSI) in DLBCL have started to gain traction, both as a means of diagnostic support but also in terms of prognostic modeling, especially by leveraging decisional networks as surrogates for the detection of molecular markers. The architecture with the most promising results utilized segmentation models, CNNs, and also assisted techniques such as AI-assisted annotations [17]. The literature reports that the aforementioned supportive measures enhanced the reproducibility and efficiency of histopathological workflows.

Work dating back to 2020 reported by Swiderska-Chadaj et al. suggested a DL system to predict MYC translocation in DLBCL using conventional H&E-stained whole-slide images (WSIs). The authors used data from 287 cases, recruited across 11 hospitals in the Netherlands, which served as the basis input data for a U-Net architecture design that was used to generate pixel-wise likelihood maps [17]. Furthermore, the data obtained was processed using a Random Forest model which designated a final classification (MYC+ vs. MYC−). The model achieved a sensitivity of 0.90–0.95 and specificity of 0.52–0.53 in internal and external validation cohorts. In a diagnostic environment which also values the judicious usage of resources, the same group concludes that this approach could reduce MYC fluorescence in situ hybridization (FISH) testing by up to 34% without compromising sensitivity [17]. As specialized FISH tests require additional logistics, a ML-aided piece of advice on who to test for this specific translocation would decrease the healthcare cost burden and also decrease the workload of the facilities which perform these tests, improving the turnaround of the results.

The digital imaging of the slides has also been suggested in predictive studies in an attempt to predict the response to the gold-standard first line therapy (R-CHOP) and also to estimate the relapse-free survival (RFS). The data for this experiment was obtained from 216 DLBCL patients selected in a Korean center, with a total of 251 slides examined. Lee et al. extracted histopathological features using a contrastive learning method (DINO—self-distillation with no labels), paired them with 54 clinical features from TabNet and modeled the input through a MIL (Multiple Instance Learning) framework [18]. The multimodal model achieved an AUROC (Area Under the Receiver Operating Characteristic curve) of 0.856 and AUPRC (Area Under the Precision–Recall Curve) of 0.961, outperforming the pathology-only model (AUROC 0.744). The prognostic validity was confirmed through Kaplan–Meier and Cox survival analyses but also involved an external validation with data from TCGA (The Cancer Genome Atlas) (*p* = 0.037), which provided a rigorous validation of the model [18]. This study proves that assisted biological tests could be translated into clinical practice for individual therapeutic planning and also as a means of treatment response assessment as an add-on to the existing imaging techniques recommended by the guides.

In another study regarding the histopathology of DLBCL, another prognostic marker was evaluated: the Ki67 proliferation index. Cristian et al. evaluated the concordance between manual and AI-assisted quantification of Ki67 in 15 cases of gastrointestinal lymphoma (13 DLBCL, NOS, and 2 HGBL). Using WSI, automated scoring was performed with QuPath on IHC stained slides, with a strong correlation between the automated method and the manual assessments (R^2^ = 0.87) [19]. This certifies the feasibility of the method for identifying high-risk cases, as elevated levels of Ki67 are associated with a poorer OS (*p* = 0.0014—reported by the authors) and a shorter PFS (also reported by the authors—*p* = 0.0028) [19].

Table 2 summarizes the risk score models based on digital pathology and optical imaging data.

### 3.3. Conventional Histopathology (Non-Digital)

The classification of DLBCL, both from a biological and from a clinical point of view, relies on traditional morphological features, immunohistochemical profiling, and documenting characteristic gene expressions. Therefore, most of the work in this particular field sought to involve ML in COO classification and also into mapping patterns which could predict the outcomes of these cases.

One study by Da Costa et al. investigated a technique meant to use immunohistochemical data from 475 DLBCL patients processed through a ML algorithm (J48). The group designed a decision-tree model that used CD10 (50% cutoff), MUM1 (5%), and FOXP1 (80%) without prior biological assumptions and achieved a kappa of 0.83 with strong concordance to gene expression profiling (GEP) of the cases studied [20]. The model performed well in terms of prognostic significance and also excelled in its prognostic capabilities for previously unclassifiable (GEP-UC) cases (PFS *p* = 0.017; OS *p* = 0.007) [20]. Worse outcomes emerged in a multivariate Cox regression, due to the non-GC class, lack of CR, and high IPI cases. Therefore, AI-assisted models can supplement traditional gene profiling in the borderline cases.

Another study involved quantitative RT-PCR data to obtain an AI powered classifier for COO subtypes in DLBCL from 143 formalin-embedded samples [21]. A set of 20 gene expression markers and 5 genes involved in the NF-κB target (a pathway renowned for its upregulation in the ABC DLBCL subtype) were measured via high-throughput qRT-PCR (Fluidigm BioMark HD system). Amongst the ML algorithms tested, the optimal one was a SimpleLogistic classifier which effectively distinguished ABC and GCB subtypes as well as returning data for the OS and NF-κB activity. Of 120 R-CHOP-treated patients, the ABC subtype was associated with a poorer prognosis, and its gene signature included elevated IRF4, CCND2, CD44, cFLIP, and CCR7 expression [21]. Therefore, a simple classifier could refine the assignment for COO case sorting and can also provide insights on the activity of various pathogenic upregulated signaling pathways.

Table 3 summarizes risk score models based on conventional, non-digital, histopathology data.

### 3.4. CT Conventional Imaging

One of the most used methods required for the management of lymphomas is computed tomography (CT), an assessment which provides essential anatomical data for staging and treatment planning and is also an indispensable tool for patient monitoring [15]. However, these imaging reports do rely heavily on the subjective visual interpretation of the operator and also on standardized morphological criteria, a step which some might argue might introduce a degree of bias, especially in the context of the complex heterogeneous masses which arise in lymphoma cases. Therefore, a need for AI-driven tools has arisen in the search for better surveillance with regard to imagistic lesions [4,5].

In the only study identified in this section, Shen et al. developed a CT radiomics model following the analysis of 1468 DLBCL patients from four hospitals in Asia, aiming to provide an automated predictive score for treatment-related cardiotoxicity for patients treated with anthracycline-based regimens. The model investigated whether an automatically calculated coronary artery calcium score (CACS) could be indicative of major adverse cardiovascular events (MACEs). Using the CACS (0, 1–100, >100) paired with logistic regression, they detected increased odds of chemotherapy-related cardiac dysfunction (CTRCD) with higher CACS values (OR 2.59 and 5.24, respectively) [22]. Moreover, an increased MACE risk was reported in both elevated CACS groups (SHR 3.73 and 7.86, respectively; *p* < 0.001), with 5-year cumulative incidences rising from 3.3% (CACS 0) to 31.1% (CACS > 100) [22]. While not focusing only on traditional markers such as OS or PFS, this tool delineates an innovative approach to supportive measures for patients with DLBCL, who could benefit from superior supportive care and better cardiological surveillance in the attempt to mitigate the undesired effects of chemotherapy.

Table 4 summarizes risk score models based on CT conventional imaging data.

### 3.5. PET-CT Imaging

The 18F-FDG PET/CT imaging test revolutionized the assessment of the disease burden and control for patients with lymphoproliferative disorders, including the subset suffering from DLBCL. The enhancement compared to traditional CT imaging techniques lies in the quantitative and spatial information on the tumor itself and also the better recognition of its activity. With the rise of AI and ML techniques, interest has also grown in leveraging these complimentary radiomic markers for the development of better predictive models during the treatment timeline. In recent years, studies have explored data extraction strategies to reveal potential biological characteristic patterns beyond the traditional SUVmax or total metabolic tumor volume (TMTV)—an accomplishment enabled by the power of complex rationalizing algorithms, which have unveiled new, subtle connections between the investigated parameters.

The focus reported was mainly on automation and also the enhancement of traditional PET scan reports. Amongst the literature we searched, we aim to report the tools listed in Table 5, which are distinct from existing work in this field.

The studies retrieved report a wide range of ML models used to predict survival outcomes and treatment responses. Notably, the considered classical models include logistic regression (LR) [24,29,30,31,32,33,34], support vector machines (SVM) [31,32,34,35,36,37], and random forest (RF) [31,32,34,37,38,39]. These methods served most commonly for binary classification while documenting treatment responses or PET positivity status. Newer studies retain the usage of deep learning architectures such as CNN [40,41,42,43], fuzzy neural networks (FNN) [31], and graph neural networks (GNN) [25], designs which extract spatial, as well as other context-specific parameters from the provided scans. The stacking method [37] or the AutoML [24,26,44] combinatorics serve the purpose of further distilling the prediction feature by integrating multiple models in the same logistic construction.

Other techniques were centered on dimensionality reduction and feature selection methods, including LASSO regression [23,26,35,37,44,45], elastic net [33], random forest ranking [38,39], and AutoML feature pipelines [26,44,45]. Future work is supported through the fact that many of the applications described were supported by open-source tools such as PyRadiomics paired with manual or semi-automated segmentation methods which have specific thresholds for selection (e.g., SUV4.0 or PERCIST criteria) [38,45].

Considering the emerging literature around AI-driven analysis of PET/CT imaging, we can infer that the improvements in this field will be reshaping the risk stratification of DLBCL patients. By combining radiomics and clinical data in upcoming risk prediction models, the iterations of future scores will likely outperform traditional tools like IPI score in predicting PFS and OS, with better expectations in terms of treatment response rates [15,37]. Worth reiterating are novel approaches such as the interpretability of spatial biomarkers (e.g., lesion-spleen distance) [28] and also models which require minimum human supervision [46], which call for a translational approach. The major challenge remains strong external validation (as many studies report only internal validation cohorts), a rigorous set of segmentation standards to be made available and used by all designed models, and a harmonization across cohorts, as independent tools have reported segmented applications of their prognostic models in what we can call the intended population for testing.

Table 6 summarizes risk score models based on PET-CT imaging data.

### 3.6. Gene Expression and Transcriptomic Profiling

Gene expression profiling (GEP) has delineated itself as a distinct field for dissecting the biological heterogeneity of DLBCL. The traditional COO classification has begun to have a more granular taxonomy with the addition of the new subgroups identified by these genetic arrays, which seek to even further refine the risk groups and impact new potential therapeutic approaches. In terms of technical data, the most promising studies applied unsupervised clustering, survival modeling, and functional pathway analyses to develop robust gene-based classifiers, validated across independent cohorts [5,54,55]. Their performances were weighted against clinically accepted and used prognostic indices.

Innovative AI methods have expanded GEP applications by uncovering biologically informed subtypes and refining prediction models. Carreras et al., contributed an algorithm designed for anomaly detection and ensemble learning (XGBoost, ANN) to identify apoptosis- and immune-related signatures (RELB) associated with outcome [56]. Another pipeline, the SurvIAE, involves a deep learning method by Zaccaria and colleagues, which integrates autoencoders, seen as an emergent trend in oncology, and multi-layer perceptrons (MLPs) to outperform R-IPI using explainable latent features [57]. Other works incorporate immune deconvolution into LASSO–Cox survival models, demonstrating that transcriptomic-immune signatures enhance outcome prediction over IPI [58]. Two other studies applied clustering and multi-algorithmic ML frameworks to model mitochondrial and programmed cell death-related expression profiles, showing strong survival associations that are validated in external datasets [59,60].

Table 7 summarizes several innovative tools for GEP.

Grouped by method, in the GEP field, AI techniques included the following: supervised learning (e.g., XGBoost, random forest, SVM) for classification and risk scoring [62,63]; unsupervised clustering (e.g., consensus clustering, SOMs) for subtype identification [60,64,65]. Hybrid pipelines such as LASSO, PSO, and PNNs were used for gene selection and survival modeling [58,66]. Deep learning frameworks like SurvIAE [57] have introduced representations of learning platforms for long-term outcome prediction.

The GEP-based AI tools shed a new light on a promising integration of novel immune classifiers as means of further subdivision of DLBCL. The cohorts described revealed superior predictive performance for personalized prognostics by using the various proposed models’ decisions [54,57,59].

Table 8 summarizes several innovative tools for GEP in DLBCL.

### 3.7. Specific Genetic Mutations

The molecular heterogeneity of DLBCL was the catalyst for the search of specific genetic markers, which could be embedded in prognostic algorithms to further refine the results. The major reason for this effort is the need for a better stratification of the patients beyond the COO and IPI frameworks. Recent ML applications involve the use of transcriptomic, mutational, and epigenetic data to characterize biologically distinct subgroups and highlight different survival scenarios. The hallmark biomarkers to-date include MYC, BCL2, BCL6 translocations, TP53 mutations, lactylation signatures, and mitochondrial gene expression, demonstrated by various authors to be of significant impact in large multicohort analyses [4,72,73,74].

Researchers have proposed several tools to refine the prognosis of such a heterogeneous disease, even at the level of specific mutations. Albitar et al. focused on a modified Naïve Bayes classifier on FFPE RNA-seq data to define four survival categories. Another group of researchers investigated oncogenic markers, such as the TP53 mutation, and implemented an algorithm pipeline consisting of 10 variables to stratify TP53-mutated DLBCL using mutation class, VAF, and structure [72]. Moreover, Zhou et al. implemented a mitochondria-based transcriptomic risk model [74] and another group of researchers proposed and tested a lactylation gene-based score, which integrates immune landscape metrics, demonstrating the importance of the tumoral microenvironmental factors apart from the transcription defects [73]. A similar approach falls partly into the multi-omics domain, documenting the importance of the stromal ambient, uncovering a stem cell-like, immune-deserted subgroup marked by ornithine decarboxylase 1 (ODC1) expression [75]. The automation enabled by ML techniques serves various purposes, ranging from gene burden scoring to metabolic and immunogenomic modeling.

Table 9 summarizes several innovative approaches to the prediction of DLBCL outcome based on genetic mutations.

Carreras and colleagues have made numerous contributions to the field by publishing studies which showcase various integrations of AI in this particular field. Their work spans across applied neural networks, decision trees, and logistic regression across gene expression and immunohistochemistry datasets to identify prognostic markers such as PD-L1, IKAROS, and Caspase-8 [77,78,79,80]. Not only did they document the aforementioned markers, but their subsequent papers proposed the use of immune-distinct signatures, the state of tumor-associated macrophages, and CSF1R patterns to define prognostic subtypes [80]. In terms of performance, the ML models achieved high OS prediction accuracy, with almost AUC 0.98 reported, and validation was performed on large datasets, demonstrating the power of layered AI pipelines [77,80].

Grouped by approach, ML techniques included supervised models like random forest, support vector machines, and logistic regression [4,81,82]; deep learning frameworks such as Bi-LSTM and multilayer perceptrons [77,81]; and unsupervised clustering tools, including iClusterPlus and SOM [83,84]. Other hybrid pipelines included LASSO–Cox, SubLymE multinomial classifiers, and gene signature selection, which further enhanced interpretability and robustness [74,85]. The diversity of AI methods which were battletested with good results once again underscores the utility and adaptability of ML models to integrate complex omics data in meaningful clinical settings.

AI-enhanced genetic profiling has substantially improved prognostic modeling in DLBCL. Even models which incorporate transcriptomics, mutational burden, metabolic signatures, and immune microenvironment profiles deliver consistent and reliable results, outperforming traditional risk assessment tools. For a wider adoption and translational use, there is still room for explainable AI frameworks and multi-omics integration, which will enable personalized therapies once clinicians have access to and have routinely worked with these new types of tools [76,83,86].

Table 10 summarizes the importance of specific gene mutations in risk score modeling in DLBCL.

### 3.8. microRNA Profiling in DLBCL

MicroRNAs (miRNAs) are small, non-coding RNA molecules that regulate gene expression post-transcriptionally, which have recently been evaluated in relation to the signaling oncogenic pathways they might reveal, disclosing dysregulation in apoptosis and immune responses. Also particular to these molecules is their stability in biofluids as well as on tissue samples, making them promising biomarkers in the setting of DLBCL as well. In recent years, ML approaches have been increasingly applied to miRNA expression data to classify molecular subtypes, forecast clinical outcomes, and identify key regulatory miRNA signatures with potential translational value. Therefore, we will try to cover the existing data on these small molecules, which were investigated for risk stratification and outcome prediction in DLBCL.

In a specialized analysis, Minezaki et al. investigated a potential diagnostic algorithm which would involve miRNA profiling in vitreoretinal lymphoma (VRL) by analyzing vitreous and serum samples from 14 VRL patients and 78 controls [89]. ML identified 17 significant miRNAs and a Random Forest model was trained to classify VRL versus controls, achieving 87.5% accuracy. miR-361-3p was the top predictive feature (AUC = 0.921). Also, the subsequent analysis of the miRNA target pathways revealed the importance of ECM–receptor interaction and IL-10 signaling [89]. ML models aided in miRNA selection for a niche subtype of DLBCL, which also shows promising diagnosis results via serum samples, based on the subset of miRNA discovered.

Nakamura et al. analyzed integrated miRNA expression profiles from two GEO datasets (GSE21848 and GSE40239), including 128 R-CHOP-treated DLBCL responders and 24 non-responders. To precisely predict the most relevant microRNAs, the study performed a selection via *p*-value ranking, stepwise selection, and Boruta (which proved to be the best selection method) and elected 11 classifiers. In individual testing, the maximum AUC of single miRNAs was modest (AUCs < 0.57), but multi-miRNA panels achieved higher predictive accuracy [90]. The authors report that a random forest Boruta-selected 36-miRNA panel scored an AUC of 0.751 in its predictive response [90]. Even if single miRNA items fail to represent a reliable biomarker in DLBCL, AUCs nearing 0.8 in relatively small clusters seem to predict the clinical use of miRNA panels for predictive purposes.

Table 11 summarizes risk score models based on microRNA profiling in DLBCL.

### 3.9. Circulating Tumor DNA (ctDNA) Analysis and Liquid Biopsy Applications

Recent studies have highlighted the utility of ctDNA as a risk stratifier, an early outcome predictor, and also as a monitoring tool in DLBCL. The current emphasis falls mostly on monitoring the plasmatic levels of this marker, often seen as a minimal residual disease (MRD) surrogate marker. However, ML data manipulation iterations have also made it a feasible tool in the early stages, with its outcome predictive properties boosting its use-case.

A study by Meriranta et al. prospectively evaluated 101 high-risk DLBCL patients to assess the prognostic utility of ctDNA. Baseline ctDNA burden, mutational profiles, and fragmentation characteristics have been obtained and analyzed, and further correlations with patient survival and MRD have been proposed. Using ML classifiers, based on 60 ctDNA-derived variables, the automated model was able to predict OS with an AUROC with values ranging from 0.75 to 0.87 [91]. Therefore, biological data derived from diagnostic samples could safely enrich the clinical decisions and satisfactorily predict the evolution of these patients, given the AUROC parameters reported.

A more targeted but equally important application of ctDNA has emerged as a biomarker in identifying central nervous system lymphoma (CNSL), particularly useful when stereotactic biopsies cannot be retrieved due to various reasons, and the imagistic workup points towards such a diagnosis. Therefore, Mutter et al. looked into a cohort of 92 CNSL patients using CAPP-Seq on plasma, CSF, and tumor samples. The first part of the experiment demonstrated the high correlation between ctDNA and the diagnosis, untreated patients presenting ctDNA in 78% of plasma and 100% of CSF samples [92]. Their work pushed forward with the design of a proof-of-concept model through a machine learning-based approach for a biopsy-free CNSL classification. The reported predictive data achieved sensitivities of 59% (CSF) and 25% (plasma), with a high positive predictive value [92]. This approach proves that AI mutation profiling tools can enhance non-invasive diagnostic tools for difficult-to-diagnose cases.

In another paper, Zhao et al. explored the predictive value of pairing the immunoglobulin heavy chain (IgH) VDJ rearrangement proportions in the ctDNA of newly diagnosed DLBCL patients. The cohort included 55 patients with two defined clonotypes: the “dominant circulating clonotype” (highest in ctDNA) and the “dominant tissue-matched clonotype” (highest in tumor tissue and found in ctDNA) [93]. The first attempt was to match it to the tissue samples’ clonotype, but the results were rather poor in predictive terms. Subsequent runs included clinical variables, the most satisfactory result involving the extranodal presence of a covariant that revealed a dominant circulating clonotype proportion ≥37% and yielded a progression prediction model with an AUC = 0.724, a sensitivity = 0.63, and specificity = 0.81 [93]. This highlights the prognostic relevance of IgH monitoring in tissue samples and reveals the promising prognostic value of a score which involves IgH tissue monitoring paired with a well-documented extra nodal involvement assessment.

Table 12 summarizes risk score models based on ctDNA in DBLCL.

### 3.10. Multi-Omics Integration

Multi-omics, an umbrella-like term, which combines data such as genomics, transcriptomics, epigenomics, proteomics, and metabolomics, offers a bundle of characteristics which display a comprehensive picture of the biology of any disease. As complex interactions are a hallmark of neoplastic cells, it is implied that the heterogeneity of cancer types relies on these interactions, thus explaining the tumor phenotype and behavior, calling for more refined classifications.

In DLBCL, there is a biological diversity that has been partially uncovered by the current classification systems, as in almost half of the cases patients either experience a poor response or relapse. Therefore, there is growing support for an approach that involves ML techniques focusing on the global picture of the pathogenic spectrum of the disease and that strive to individualize the cases based on markers that are not frequently documented through an emphasis on pathway signaling changes, which do occur during the life-span.

As early as 2003, Futschik et al. described the development of a hierarchical modular model which integrates gene expression microarray data (the omics part) and the clinical IPI score to predict the outcomes in DLBCL. From a technical standpoint, the IPI scores were modeled using a Bayesian classifier, while the microarray data was processed using evolving fuzzy neural networks (EFuNN); the work was based on a preexisting lot of 58 patients [94]. An interesting feature of the comparison between the microarray results and the IPI score was the fact that both seemed to be independent in terms of prediction, even in pathways that should have been shared between them. Therefore, the idea was sparked to imagine a combined mode using a hierarchical modular model, which, when tested, achieved a prediction accuracy of 87.5%, outperforming either modality alone [94].

In another study, Mosquera Orgueira et al. examined 481 DLBCL patients from a UK cohort, using transcriptomic features via the LymForest-25 algorithm paired with IPI derived data and mutational identified clusters. Their techniques suggested the use of PCA and Cox regression in order to constitute a multi-omics survival model with internal bootstrapping [95]. They also performed the tests on different age groups, obtaining the best performance in the <70 year subgroup, achieving AUCs of 0.82 at 5 years and over 0.81 from 0.5 to 2 years [95]. This strongly suggests that adding molecular data to the traditional prognostic score would boost its predictive performance.

The use of LymForest-25 is highlighted also in another study led by the same author that looked into a more specific scientific question, the benefit of adding a proteasome inhibitor to the old R-CHOP regimen. Therefore, after training the model on a 469 patients cohort treated with R-CHOP, they tried to predict the PFS for the subgroup with the addition of bortezomib (RB-CHOP) [96]. The prediction yielded a significant 30% reduction or death for the DLBCL patients at higher molecular risk [96].

Further studies tried to improve the existing models; one of them even describes the development of an empirical Bayes version of Bayesian additive regression trees (EB-coBART) to predict 2-year PFS in DLBCL. A model composed of 101 uniformly treated DLBCL cases, with 140 covariates factored in, demonstrated a superior predictive performance in the EB-coBART model with a C-index of 0.714 [97]. This data reveals that robust multi-omics models can be constructed to demystify the biological prognostic in this subtype of lymphomas.

Another distinctive approach was applied in the paper by Ouyang et al., who applied ML strategies to identify novel cell death pathways such as cuproptosis in DLBCL. They used featured selection techniques (Random Forest, LASSO, Boruta, Univariate Filtering) and a Transformer-based deep learning model to identify lncRNAs from transcriptomic data across five large DLBCL datasets. The final group included 831 patients and the predictive model created for OS prediction achieved AUCs of 0.79–0.83 in internal validation and 0.66–0.69 in external cohorts. Also, the study highlights lncRNA MALAT1 as a consistent key prognostic factor for cell proliferation in DLBCL, with subsequent reductions in cell division in DLBCL cell lines with MALAT1 knockdown, raising questions about its potential therapeutic role as well [98].

Table 13 summarizes risk score models based on multi-omics in DLBCL.

### 3.11. Superiority of AI/ML over Traditional Prognostic Indices

While the studies analyzed datasets that differed across clinical, imaging, histopathologic, and multi-omics dimensions, there were nonetheless common, recurring, high-impact predictors identified across the various, independent AI and ML frameworks. In order to build a useful, translational perspective, a distillation of the most common parameters would illustrate the most promising data types to prioritize for potential prognostic feature extraction. Accordingly, we extracted and ranked the variables that were most frequently or successfully consolidated from the studies we reviewed—those which consistently emerged as leading parameters within the various feature selection AI algorithms (e.g., LASSO, Random Forest ranking, AutoML pipelines) or that were retained as independent variables within the context of multivariable survival analysis. The table below (Table 14) presents the ***top 13 prognostic indicators*** that exhibited consistent predictive value for **PFS** and **OS** in DLBCL, including their data source, key studies, and the extent to which they contributed to the predictive model.

Integrating evidence across different fields reveals a shared ‘core’ set of prognostic indicators regardless of the dataset or model architecture used. These include age; LDH; ECOG performance status; extranodal involvement; and radiomic features such as total metabolic tumor volume (TMTV) and dissemination indices (Dmax and spleen-referenced distances) as well as molecular profiles (CAF, TP53, and mitochondrial gene panels). These indicators have been confirmed repeatedly in different studies, contributing to the performance of models scoring C-index or AUC values of 0.74 to 0.87 as opposed to the older IPI-based scores (0.60–0.70) [8,23,26,33,55,57,72,74]. These findings indicate that hybrid models in AI should include these variables as primary components. This synthesis thus connects distinct methods with practical goals and concentrates evidence-based DLBCL AI research on reproducibility and clinical integration.

### 3.12. Model Interpretability and the Path to Clinical Translation

An important factor in the clinical translation of AI prognostic models is the need for a level of transparency. The inability to interpret such models remains a barrier to widespread adoption. In healthcare environments, explainability determines trust. A physician will want to have a full understanding of the rationale behind the model’s high-risk prediction for a particular patient before any treatment is based on such a prediction. Our analysis points out that the most advanced AI prognostic models, especially those based on clinical or genomic data, have utilized interpretable machine learning approaches. Notably there was a tendency towards models such as LASSO, which penalizes the Cox regression and therefore aids in the selection of key features for patient stratification [13,16]. Another widely used framework was represented by XGBoost or other similar ensemble methods which explain feature importance, thus determining the selection of the most vital markers to be used in the stratification process (see Table 14) [10,11,14]. This approach captures salient prognostic markers such as age, ECOG, and particular mutational profiles [10,13]. In stark contrast, the literature lacks data on the adoption of advanced models. In particular, the use of SHAP (SHapley Additive exPlanations) or LIME (Local Interpretable Model-agnostic Explanations) to explain the prediction of a model for an individual patient is remarkably absent for models considered to be part of post hoc XAI class.

Future research should focus beyond mere performance metrics and take into consideration how AI-derived insights can be rendered clinically actionable and understandable, thereby facilitating a seamless transition from research to real-world workflow integration.

## 4. Discussions, Future Perspectives

### 4.1. Synthesis of Findings and Domain-Specific Trends

This scoping review underscores the rapidly evolving landscape of ML in prognostic modeling for DLBCL. Our work focused on an in-depth collection of the expanding literature in this field, mainly documenting ML approaches in relation to the investigated markers, while also synthesizing the outcomes obtained by each individual experiment.

We underline a consistent finding across the scouted domains in the superior discriminatory capacity of ML models. In PET/CT imaging, CNNs operating on maximum intensity projections (MIPs) or radiomics-enhanced input data, with impressive AUCs > 0.8 for PFS prediction or treatment failure, data which is independently validated [26,27,30,32,33,36,44,99]. It has been demonstrated that even small changes in quantitative data such as the MTVrate (MTV lesion/total volume), empowered robust logistic regression models with strong stratification capability, transforming available data into valuable input for classifiers [33]. Therefore, we can safely say that ML can detect subtle spatial or volumetric patterns and extract data indicative of aggressive biology earlier than visual reads or SUVmax alone.

Survival endpoints have always been an area of interest in terms of prognostics. The review of the existing literature uncovered a pattern which emerged from linking specific input parameters and superior model performance for PFS and OS prediction. Radiomics-based PET/CT models which rely on patient data regarding the total metabolic tumor volume (TMTV), lesion dissemination indices, and metabolic tumor volume rate have consistently reported C-index or AUC values ranging between 0.75 and 0.80, performances which yield slightly superior results to IPI-based benchmarks (reported C-indices/AUCs values of 0.60–0.72) using data derived from the independent cohorts mentioned by the authors [23,26,27,28,33,37,44,48].

Another sought endpoint of clinical significance, PFS prediction, was investigated in some of the reviewed papers. The most appealing results came from convolutional and ensemble learning frameworks, where spatial metrics such as spleen-referenced dissemination captured the strongest discriminatory capacity (sDmax HR ≈ 11–12; ΔC-index +0.05–0.07)—capturing early metabolic and spatial heterogeneity of disease [28,37,51].

OS-oriented models have demonstrated the highest prognostic accuracy by using data derived from multi-omics or transcriptomic datasets. Amongst the input variables, studies used either complex molecular signatures (gene-expression patterns, immune microenvironment composition) or TP53 mutation-aware profiles, to classify and decode the biological heterogeneity of long-term outcome trends and treatment resistance patterns. The best performing models, which achieved C-index values ranging from 0.78 to 0.87, include the CAF-related transcriptomic signature [55], the autoencoder-based SurvIAE model [57], and mitochondrial or TP53-integrated risk scores [59,72,74,75]. These C-index values above 0.7 mark these approaches as superior to the IPI or NCCN-IPI models (which have a mean performance lower than 0.7), highlighting the fact that processing high-dimensional molecular data through interpretable ML architectures is positioned to enable refined survival predictions, calling for further investigations of treatment intensity tailoring according to biological lymphoma aggressiveness grading.

Classical clinical variables such as age, serum LDH, ECOG performance status, and extranodal involvement maintained a strong and independent prognostic value amongst the described AI-based models, irrespective of the model complexity or data integration workflow [8,13,14,15,33]. The large registry-based studies by Biccler et al., which analyzed a cohort of over 4000 DLBCL cases through both random survival forest and stacked Cox models, reported the consistent predictive value of the age and ECOG parameters (overall model C-indices of 0.74–0.76) that were superior to the IPI benchmark (C-index ≈ 0.70) [8].

Another reinforcing signal of the robust predictive value of clinical parameters was demonstrated by Shen et al., who reported that the individual variables of age, WBC count, hemoglobin, and LDH were retained as top-ranking coefficients in their penalized Cox (LASSO) model (C-index of 0.704 and AUC 75.8%), outperforming the conventional established scores [13]. Furthermore, a SEER-based cohort of 869 patients analyzed by Zhao et al. highlighted the age and extranodal disease parameters’ impact on survival, using integrated XGBoost models (AUCs > 0.80) for 5-year and 10-year mortality prediction [14]. Additionally, another Cox–XGBoost model by Zhu et al., using LDH and time to relapse > 12 months as input data, reached AUCs 0.863–0.898 for OS and 0.769–0.784 for PFS [15]. Even in imaging-enhanced models such as that proposed by Czibor et al., which used additional PET/CT features, biological markers, such as LDH, maintained their role as top-ranked predictor variables (AUC of 0.83 for 24-month PFS) [33].

Collectively, we cannot fail to notice the robust emerging data which indicate that various AI and ML framework designs re-validate established clinical markers. Rather than rendering these biomarkers obsolete, the published papers highlight these markers as core, high-impact variables which maintain strong discriminative power as a backbone for improved scoring systems alongside radiomic or genomic features. Therefore, we have to emphasize that there is a strong need for hybrid models that leverage accessible, low-cost clinical parameters paired with advanced computational input-processing models, targeting superior classification strategies which do not replace the existing ones, but complement the previously designed and well-used performance indices.

### 4.2. Challenges, Knowledge Gaps, and Future Directions

The same trend has been seen and implemented successfully in other hematologic malignancies. A pivotal AML study by Eckardt et al., developed and validated nine ML models trained on clinical, cytogenetic, and molecular features from 1383 intensively treated patients, with impressive AUROC values of 0.77–0.86 for complete remission and 0.63–0.74 for 2-year survival, exceeding conventional ELN-based scores [100]. Also, other researchers inferred, from peripheral blood smears, an early relapse predictor for AML using a transformer-based deep learning model, outpacing the existing clinical risk groups [101]. The concordance of these results depicts that multi-step ML pipelines, which are still under development, can already enhance risk stratification across hematologic cancers.

In DLBCL, there is a diversity of input modalities. Data can be derived from digital histopathology, gene expression, methylation, and ctDNA mutation profiles, streams that raise the opportunity for ML models to integrate disparate signals into unified risk scores. The increasing number of data inputs perfects the prediction capabilities of the ML model; multi-omics tools such as LymForest-25 prognostic signatures outperformed traditional indices and often revealed immune, stromal, or metabolic features predictive of poor outcomes [95]. A more targeted approach involves a narrower path but is extremely lucrative in extracting biologic drivers of risk, such as through the use of MYC signaling or T-cell exclusion [83,102].

Moving a step backward, the performance and interpretability of the ML-based models raises their use into the domain of clinically actionable decision tools. One example is the identification of patients with high-risk features (e.g., high lesion count or immune-desert transcriptomes) at baseline, which could guide early trial enrollment, up-front CAR-T referral, or intensified immunochemotherapy. As a parallel, AML random forests and logistic models using molecular data such as NPM1, FLT3-ITD, and TP53 mutations reshaped induction strategies once they were uncovered [100]. Therefore, we anticipate that translational use in DLBCL might have a similar impact if integrated into pre-treatment workflows.

### 4.3. Strengths and Limitations of the Scoping Review Methodology

However, we should not fail to acknowledge some challenges which still have to be mitigated. During our research, we encountered a heterogeneity in sample sizes, validation methods, and even in model transparency reporting amongst papers. We have noticed that few studies accounted for real-world deployment constraints, mostly relying on data derived from preexisting retrospective cohorts, with few mentions or design-wise notes related to computational cost, sequencing availability, or the need for model explainability, as models are sometimes regarded as black-boxes by clinicians, with their good outcomes but obscure workflows making clinicians skeptical of relying on them [6]. A long-awaited harmonization of reporting standards, integrated data feeding systems, and clear work concepts would improve clinical workflow integration, providing confidence amongst its users.

We also have to report some clinical limitations of this review. As this paper represents a scoping review, a formal risk of bias assessment was not performed, in line with the PRISMA-ScR guidance considering the heterogeneous nature and primary objective of the included studies, which was a systematic categorization rather than a synthesis of effect estimates [7]. Therefore, we focused on mapping the breadth and diversity of machine learning applications in DLBCL. Also, we have to reiterate the vast subdomains included, with highly heterogeneous sample sizes, data modalities, and model architectures, which precluded direct comparison or quantitative synthesis between all the articles. A third hindrance would be the high number of studies that lacked external validation, thus bringing the performance of some models in uncontrolled settings into question. Also, there might be a publication bias, especially against reporting negative or poorly performing ML models, which may thus be underrepresented in the literature.

## 5. Conclusions

To conclude, this review highlights that ML methods have the potential to predict outcomes in DLBCL earlier and more accurately than traditional approaches. The existing IPI score exhibited a satisfactory predictive power, but many of the reported models demonstrated enhanced discriminative features, even in patients which did not display a clear biological heterogeneous phenotype as per current classification standards. ML techniques uncovered non-obvious risk patterns, especially in multi-omics cohorts, which are not yet fully described in the literature. The strength of all the studies resided in the constant outperformance of the IPI score as a prognostic metric, and several studies also hinted at promising applications for personalized therapies in some cohorts. It was unanimously mentioned that all the studies tried to emphasize the use-case of decisional structures in accelerating the time-to-intervention in high-risk patients, a goal which is also intensely investigated in other hematological malignancies.

## Figures and Tables

**Figure 1 medsci-13-00280-f001:**
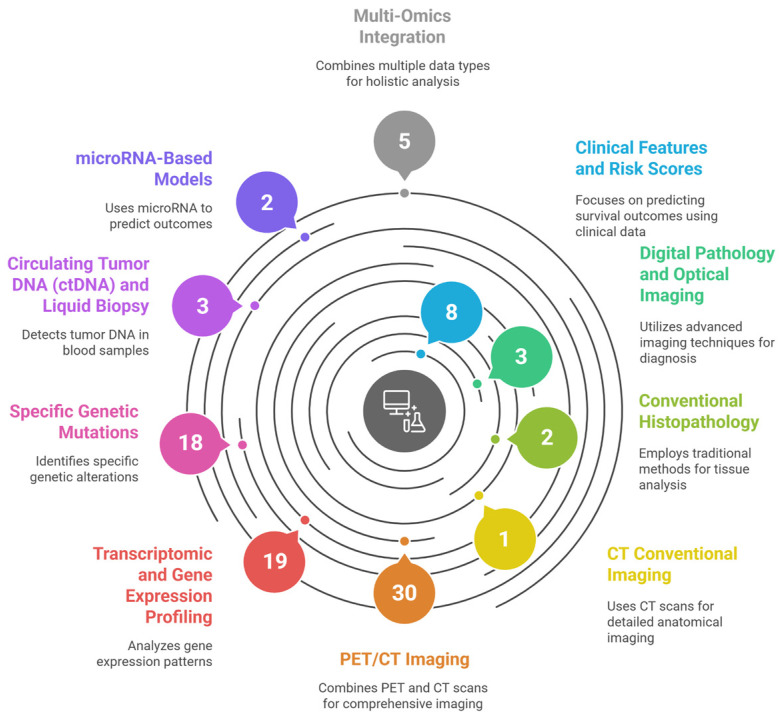
Graphical abstract—*Artificial Intelligence for Risk Stratification in Diffuse Large B-Cell Lymphoma: A Systematic Review*, illustrating the systematic categorization of 91 studies applying Artificial Intelligence and Machine Learning for prognostic modeling in Diffuse Large B-Cell Lymphoma (DLBCL). The figure maps the current research landscape by grouping studies into ten distinct data modalities and computational approaches identified in the review’s results, including *Clinical Features and Risk Scores*, various *Imaging Techniques* (PET/CT, CT, and Digital Pathology), and diverse *Omics data* (Transcriptomic, Specific Genetic Mutations, ctDNA, microRNA, and Multi-Omics Integration). This figure was generated using the AI tool napkin.ai version 3, based on the “Overview of Included Studies” (Section 3) of this systematic review. AI, artificial intelligence. DLBCL, diffuse large B-cell lymphoma.

**Figure 2 medsci-13-00280-f002:**
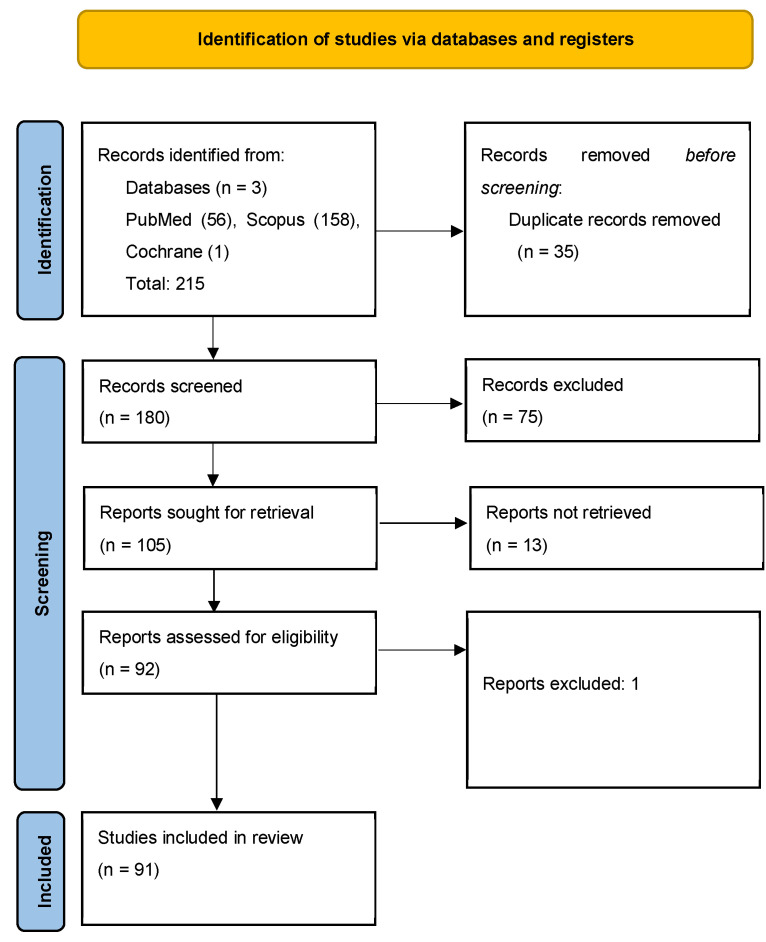
PRISMA-SCR flow diagram outlining the study selection process for the scoping review on the implications of AI/ML to uncover prognostic and predictive factors in DLBCL [7].

**Table 1 medsci-13-00280-t001:** Clinical features and risk score modeling—summary table.

First Author (Year)	Data Modality	Dataset Size/Source	Input Features	AI/ML Method	Outcome Predicted	Model Performance	Validation Strategy
Biccler (2018a) [16]	Clinical	4420 DLBCL patients/Danish National Lymphoma Registry	Age, stage, LDH, ECOG, extranodal sites, + up to 20 clinical variables	RSF, penalized Cox, CPH	Overall Survival	tAUC > IPI (exact value not provided)	10-fold cross-validation
Biccler (2018b) [8]	Clinical	5173/Danish + Swedish National Registries	Age, ECOG, LDH, stage, albumin, etc.	Stacked model (CPH, AFT, parametric)	Overall Survival	C-index 0.756 (DK); tAUC ~0.75; IBS 0.131 vs. IPI 0.150	10-fold CV + External (SE cohort)
Fan (2021) [9]	Clinical + Molecular	510 training/530 external validation	Age, stage, LDH, β2-microglobulin, extranodal sites, genetic mutations	Logistic regression + ML classifiers (LR, NB, RF, SVM, FFNN)	Overall Survival	C-index 0.71	External validation cohort
Kim (2024) [10]	Clinical + IHC	134 DLBCL/single-center cohort	NCCN-IPI, stromal FOXC1, tumor pERK1/2	FastSVM, RF-Survival, Survival Tree, XGBoost, GBoostSurv, Bagging SurvTree	Time to death	C-index 0.801 (NCCN-IPI + FOXC1 + pERK1-2, *p* = 0.030)	Internal comparison
Qin (2024) [11]	Clinical (SEER data)	836 pediatric DLBCL patients (2000–2019)	Age, sex, Ann Arbor stage, surgery, chemotherapy, radiotherapy, systemic therapy, diagnosis delay	XGBoost, Cox regression, Generalized Cox (NPH, NLL, NPHNLL)	Overall survival	XGBoost AUC: 0.892 (train), 0.889 (validation)	Internal validation (7:3 split)
Samarina (2019) [12]	Clinical + IHC (Hans)	81 DLBCL/single-center (Russia)	IPI risk group, GCB/non-GCB subtype	CART	Overall survival	2-yr OS: 100% (low risk), 74% (intermediate), 46% (high); median OS 25 mo (high)	Internal comparison only
Shen (2022) [13]	Clinical	1211/7 Chinese centers	Demographic, ECOG, Ann Arbor, CBC indices, albumin, cholesterol, GCB subtype	LASSO (penalized Cox) vs. Random Forest	Overall survival	LASSO AUC 75.8%, C-index 0.704 (train)	Internal split (training vs. validation)
Zhu (2025) [15]	Clinical + PET/CT	28/The First Affiliated Hospital of Zhejiang CMU	CD5+, LDH, ALB, β2-MG, initial response, time to relapse, anemia, stage, etc.	RSF, GBM, Cox-XGBoost, Stepwise Cox	OS and PFS (1–3 yr)	OS AUC (test): 0.863–0.898; PFS AUC (test): 0.769–0.784	Internal (train/test split)

Legend: AFT, accelerated failure time; ALB, albumin; ANN, artificial neural network; AUC, area under the curve; β2-MG, beta-2 microglobulin; CART, classification and regression tree; CBC, complete blood count; C-index, concordance index; CMU, China Medical University; CPH, Cox proportional hazards; CT, computed tomography; CV, cross-validation; DLBCL, diffuse large B-cell lymphoma; ECOG, Eastern Cooperative Oncology Group; FFNN, feedforward neural network; FOXC1, forkhead box C1; GBM, gradient boosting machine; GBoostSurv, gradient boosting survival model; GCB, germinal center B-cell; Hans, Hans algorithm for COO subtyping; IHC, immunohistochemistry; IPI, International Prognostic Index; IBS, integrated Brier score; LDH, lactate dehydrogenase; LR, logistic regression; ML, machine learning; NB, naïve Bayes; NCCN, National Comprehensive Cancer Network; NCCN-IPI, NCCN version of the IPI; NLL, non-linear link; NPH, nonproportional hazards; NPHNLL, nonproportional hazards with non-linear link; OS, overall survival; PFS, progression-free survival; PET, positron emission tomography; pERK1/2, phosphorylated extracellular signal-regulated kinases 1 and 2; RF, random forest; RSF, random survival forest; SEER, Surveillance, Epidemiology, and End Results Program; SVM, support vector machine; Stepwise Cox, stepwise Cox regression; tAUC, time-dependent area under the curve; XGBoost, extreme gradient boosting.

**Table 2 medsci-13-00280-t002:** Digital pathology and optical imaging—summary table.

First Author (Year)	Data Modality	Dataset Size/Source	Input Features	AI/ML Method	Outcome Predicted	Model Performance	Validation Strategy
Swiderska-Chadaj (2020) [17]	Digital pathology (H&E)	287 WSIs/11 hospitals (Netherlands)	H&E image morphology	U-Net CNN + Random Forest	MYC rearrangement	Sensitivity 0.90–0.95; Specificity ~0.52	Internal + external (WSI split)
Lee (2024) [18]	Digital pathology + clinical	251 WSIs from 216 cases/single center (Korea)	WSI morphology (contrastive learning) + clinical (TabNet)	DINO + MIL + TabNet + UMAP (no tumor annotation)	Response to R-CHOP, RFS	AUROC: 0.744 (Histo), 0.856 (Multimodal); AUPRC: 0.935/0.961; log-rank *p* = 0.041/0.026/0.037	5-fold CV + external (TCGA survival)
Cristian (2023) [19]	Digital pathology (IHC)	15 GI lymphoma cases (13 DLBCL, 2 HGBL)/single center	Ki67 IHC images	QuPath (automated quant)	OS and PFS (via Ki67 score)	R^2^ = 0.87; OS *p* = 0.0014; PFS *p* = 0.0028	Manual vs. AI agreement; survival stratification

Legend: AUROC, area under the receiver operating characteristic curve; AUPRC, area under the precision–recall curve; CNN, convolutional neural network; CV, cross-validation; DINO, self-distillation with no labels; DLBCL, diffuse large B-cell lymphoma; GI, gastrointestinal; H&E, hematoxylin and eosin; HGBL, high-grade B-cell lymphoma; IHC, immunohistochemistry; MIL, multiple instance learning; OS, overall survival; PFS, progression-free survival; QuPath, open-source bioimage analysis software version 0.4.3; R-CHOP, rituximab, cyclophosphamide, doxorubicin, vincristine, prednisone; RFS, relapse-free survival; TabNet, tabular data deep learning model; TCGA, The Cancer Genome Atlas; UMAP, Uniform Manifold Approximation and Projection; U-Net, convolutional network for biomedical image segmentation; WSI, whole-slide image.

**Table 3 medsci-13-00280-t003:** Conventional histopathology (non-digital)—summary table.

First Author (Year)	Data Modality	Dataset Size/Source	Input Features	AI/ML Method	Outcome Predicted	Model Performance	Validation Strategy
Da Costa (2018) [20]	Conventional histopathology	475 patients (Visco-Young dataset)	IHC markers: CD10, MUM1, FOXP1	J48 decision tree	Cell of origin, OS, PFS	Kappa = 0.83; OS *p* = 0.007; PFS *p* = 0.017	10-fold CV (internal); Kaplan–Meier; Cox model
Xue (2015) [21]	Histopathology (qRT-PCR from FFPE)	143 DLBCL patients (120 R-CHOP treated)	20 COO classifier genes, 5 NF-κB target genes	SimpleLogistic	COO subtype, OS	OS *p* = 0.043; elevated NF-κB expression in ABC	External (public GEP datasets) + OS stratification

Legend: ABC, activated B-cell-like; COO, cell of origin; CV, cross-validation; DLBCL, diffuse large B-cell lymphoma; FFPE, formalin-fixed paraffin-embedded; FOXP1, forkhead box protein P1; GEP, gene expression profiling; IHC, immunohistochemistry; J48, C4.5 decision tree classifier; MUM1, multiple myeloma oncogene 1; NF-κB, nuclear factor kappa-light-chain-enhancer of activated B cells; OS, overall survival; PFS, progression-free survival; qRT-PCR, quantitative reverse transcription polymerase chain reaction; R-CHOP, rituximab, cyclophosphamide, doxorubicin, vincristine, prednisone.

**Table 4 medsci-13-00280-t004:** CT conventional imaging—summary table.

First Author (Year)	Data Modality	Dataset Size/Source	Input Features	AI/ML Method	Outcome Predicted	Model Performance	Validation Strategy
Shen (2023) [22]	CT imaging (non-gated)	1468 DLBCL patients/4 hospitals (Asia)	AI-derived CACS	Automated CACS scoring + regression models	CTRCD and MACEs	OR for CTRCD: 2.59–5.24; SHR for MACE: 3.73–7.86; all *p* < 0.001	Internal retrospective stratification + Fine–Gray competing risk models

Legend: AI, artificial intelligence; CACS, coronary artery calcium score; CT, computed tomography; CTRCD, cancer therapy-related cardiac dysfunction; DLBCL, diffuse large B-cell lymphoma; MACEs, major adverse cardiovascular events; OR, odds ratio; SHR, subdistribution hazard ratio.

**Table 5 medsci-13-00280-t005:** Innovative PET-CT AI tools in DLBCL.

Tool Name	Innovative Features
PET automated region segmentation—PARS (Capobianco 2021) [23]	A CNN-based segmentation method which classifies lesions for automated TMTV quantification
AutoGluon (Zhao 2025) [24]	An automated machine learning tool which generates *radscore* and provides treatment response prediction
Graph Neural Network with cross-attention fusion (Thiéry 2024) [25]	A graph modeling tool which seeks to examine lesion-level patterns and cross-checks with clinical and imaging data
PET-fusion DFR-signature (Chen 2024) [26]	A tool leveraging deep features of combined PET/CT images -> integrated into an AutoML survival prediction model
MIP-CNN (Ferrandez 2024) [27]	A CNN centered on maximum intensity projection images (MIPs) without needing a segmentation step
Spleen-referenced radiomics (Girum 2024) [28]	A model based on dissemination metrics involving spleen anatomy distances with an OS/PFS prediction model

Legend: AutoML, automated machine learning; CNN, convolutional neural network; CT, computed tomography; DFR, deep feature radiomics; DLBCL, diffuse large B-cell lymphoma; MIP, maximum intensity projection; OS, overall survival; PARS, PET automated region segmentation; PET, positron emission tomography; PFS, progression-free survival.

**Table 6 medsci-13-00280-t006:** PET-CT imaging—summary table.

First Author (Year)	Data Modality	Dataset Size/Source	Input Features	AI/ML Method	Outcome Predicted	Model Performance	Validation Strategy
Capobianco (2021) [23]	PET/CT	250 patients (R-CHOP-treated DLBCL)	Radiomic features from baseline PET	LASSO + Cox regression	OS, PFS	C-index 0.74 (OS), 0.72 (PFS)	10-fold cross-validation
Carlier (2024) [29]	PET/CT + Clinical	545 patients (GAINED study)	PET radiomics + clinical variables	Logistic regression + penalized Cox	2y PFS	C-index gain minimal with PET radiomics	Internal validation
Ferrandez (2023) [43]	PET/CT	1132 patients from 5 trials	Maximum-intensity projection (MIP) images	CNN (ResNet)	2y PFS	CNN AUC = 0.66 vs. IPI AUC = 0.60	External validation (5 trials)
Ferrandez (2024) [27]	PET/CT	177 high-risk DLBCL	PET radiomics	7 × 7 ML cross-combinations (e.g., LASSO-LASSO)	Mid-treatment response, PFS, OS	RadScore AUC > clinical models	Cross-validation
Frood (2022) [47]	PET/CT	296 patients	Radiomic features (GLSZM, etc.)	Ridge regression	2y EFS	AUC = 0.75 (val), 0.73 (test)	Split-sample validation
Girum (2022) [48]	PET/CT	382 patients from REMARC and LNH073B	MIP-based surrogate TMTV and Dmax	AI segmentation algorithm	PFS	HRs for sTMTV and sDmax ~11–12	External validation across trials
Girum (2024) [28]	PET/CT	282 patients from REMARC	SpreadSpleen, Dspleen, sDspleen + TMTV + IPI	AI segmentation + multivariate Cox	PFS, OS	Improved C-index with spleen-relative features	Internal validation
Huang (2022) [46]	PET/CT	147 DLBCL patients (Henri Becquerel Center)	Weakly labeled and fully labeled PET/CT	3D V-Net + ASPP, Cosine similarity	Tumor segmentation	DSC = 71.47% (avg WS), 75.21% (FS_120)	Experiments: FS_60, WS_60_60, FS_120
Jemaa (2022) [49]	PET/CT + Clinical	1268 patients from GOYA trial (NCT01287741); training: 846; hold-out: 422	Deep learning-extracted PET metrics (TMTV, lesion distribution, CNS risk) + IPI components	Fully automated AI-based segmentation and prediction pipeline	4-year PFS, OS; 2-year CNS relapse risk	PFS HR 1.87 (95% CI: 1.31–2.67); OS HR 2.16 (95% CI: 1.37–3.40); C-index 0.59 vs. 0.55 (vs. IPI); 17.1% 2-year CNS relapse risk	Stratified train–test split (33% hold-out balanced by treatment, IPI, PFS/OS events, TMTV); model selection via univariate Cox PH + LASSO + 5-fold CV; final model tested on hold-out set using C-index and AUC
Jesus (2021) [30]	PET/CT Imaging	120 patients (44 FL, 76 DLBCL); 348 lesions	79 PET radiomics, 51 CT features, 6 shapes	Gradient Boosting (vs. LR, AdaBoost, XGBoost)	Discrimination of FL vs. DLBCL	AUC = 0.86, Accuracy = 80% (vs. SUVmax AUC = 0.79)	Internal testing with *p*-value comparison to baseline
Jiang (2022) [50]	PET/CT Radiomics	140 PGI-DLBCL patients, single-center (pre-treatment FDG PET/CT)	1421 radiomic features from PET (reduced to 5 via ensemble feature selection); combined with metabolic metrics and clinical risk factors	SVM classifier (radiomics signature); multivariate Cox regression (combined model)	Progression-free survival (PFS), overall survival (OS)	Radiomics signature alone significantly associated with OS and PFS (*p* < 0.05); combined model: PFS C-index = 0.831, OS = 0.877 (validation)	Split into training and validation sets; performance tested with Cox model and decision curve analysis (DCA)
Jiang (2022) [51]	PET Radiomics	383 DLBCL patients from two medical centers (2011–2019)	3 types of PET radiomics features, metabolic metrics, clinical risk factors	7 ML classifiers via cross-combination, Cox regression	PFS, OS	Combined model C-index (PFS: 0.801, OS: 0.807); external C-index (PFS: 0.758, OS: 0.794)	Internal and external validation (training and validation cohorts); DCA, calibration curves
Jing (2023) [52]	PET/CT Radiomics	201 DLBCL patients (China)	1328 radiomics features from baseline 18F-FDG PET/CT; clinical and metabolic variables	LASSO + Cox regression	2-year progression-free survival (PFS), 5-year overall survival (OS)	Radiomics-only model: C-index PFS = 0.732, OS = 0.759; AUC PFS = 0.768, OS = 0.767; outperformed clinical and metabolic models	Internal validation using hold-out test set
Chang (2023) [31]	PET/CT + Clinical data	122 patients (Taiwan)	IPI, laboratory parameters, MTVsum (from baseline FDG-PET/CT)	Logistic regression, Random Forest, SVC, DNN, FNN	3-year PFS and 3-year OS (binary classification)	Best accuracy: PFS—71% (SVC, DNN); OS—76% (DNN)	10-iterated fivefold cross-validation with shuffling
Chen (2024) [26]	Fused PET/CT + Clinical	369 patients (2 centers: Nanjing and Sichuan)	DFR-signature (1000 deep features from fused PET/CT) + NCCN-IPI	AutoML (AutoGluon)	PFS and OS	C-index: PFS—0.784 (train), 0.739 (val); OS—0.831 (train), 0.782 (val)	10-fold CV; internal validation
Chen (2024) [53]	PET Radiomics	177 high-risk DLBCL patients from 2 centers (2012–2022)	110 handcrafted PET features + SUVmax, MTV, TLG	7 × 7 ML cross-combinatorial pipeline (49 models); LASSO–LASSO for RadScore	Mid-term treatment outcome, PFS, OS	AUC (combined model): 0.846 (train), 0.724 (val); RadScore: OR = 7.17; PFS HR = 2.17, OS HR = 2.14	Random 70:30 split into training (n = 123) and validation (n = 54); KM and Cox analysis
Czibor (2024) [33]	PET Radiomics + Clinical	50 DLBCL patients (baseline [18F]FDG-PET/CT)	MTV, novel MTVrate, LDH, 1st–3rd order texture features from largest lesion VOI	Logistic regression (ML-based prognostic model)	24-month PFS	AUC = 0.83 (ML model); best individual: MTVrate AUC = 0.74	ROC analysis; log-rank; subgroup stratification
Detrait (2024) [32]	PET/CT + Clinical + Treatment data	130 DLBCL patients (2017–2022)	Demographics, disease features (Ann Arbor stage, IPI, CMV/EBV), treatment type, early PET-CT after 2 cycles	5 ML models: Naïve Bayes (categorical), XGBoost, RF, SVM, Logistic Regression	Primary refractory disease	Naïve Bayes: AUC = 0.81, Accuracy = 83%, F1 = 0.82, FPR = 10%; others: XGBoost AUC = 0.74, RF = 0.67	Performance metrics reported on validation set; no external test cohort
Jing (2025) [45]	PET/CT Radiomics + Clinical	126 DLBCL patients with extranodal involvement (ENI)	1328 PET/CT radiomics features (PyRadiomics), SDmax, clinical data	LASSO–Cox for feature selection; Cox regression for model building	PFS and OS	Combined model C-index: 0.724 (PFS), 0.842 (OS); radiomics-only: 0.704/0.744; clinical-only: 0.615/0.792	Internal validation with AIC comparison; bootstrap resampling; KM + log-rank for survival analysis
Kuker (2022) [40]	PET/CT (MTV quantification)	100 newly diagnosed DLBCL patients (Alliance/CALGB 50303 trial)	Automated segmentation of physiologic structures using CT + FDG-PET overlay	Deep convolutional neural network (CNN)-based segmentation (fully automated MTV pipeline)	Agreement with manual MTV and SUVmax quantification	Pearson r ≈ 0.98; ICC = 0.98 vs. both readers; minimal bias on Bland–Altman plots	Internal validation against 2 expert readers (manual MTV assessment); ICC, Pearson r, Bland–Altman analysis
Liu (2024) [34]	Brain MRI Radiomics (CE-T1WI, FLAIR)	102 PCNSL patients (31 BCL6+, 71 BCL6−), single-center	Radiomics from VOI_tumour and VOI_peritumour (CE-T1WI, FLAIR)	LASSO for feature selection; ML classifiers: LR, RF, SVM, KNN	BCL6 gene rearrangement (binary classification)	Best AUC (logistic regression): 0.935 (train), 0.923 (validation)	7:3 train/validation split; univariate + multivariate logistic regression
Luo (2024) [36]	PET/CT + Clinical	187 newly diagnosed DLBCL patients	Imaging features (PET/CT radiomics), clinical features (age, LDH, ECOG, etc.)	Multi-view learning: SVM with kernel canonical correlation analysis (SVM-2 KCCA)	Prognosis/survival classification	AUC = 0.921; Accuracy = 96.9%; F1 = 92.8%; Sensitivity = 90.9%	Train/test split with comparison to 3 other MVL models; metrics: AUC, accuracy, F1, G-mean
Pinochet (2021) [41]	PET/CT (TMTV segmentation)	119 DLBCL patients (research cohort) + 430 mixed-cancer patients (clinical cohort)	Baseline PET scans; manual vs. automated TMTV using CNN-based PARS prototype	Convolutional Neural Network (CNN) (PARS, Siemens Healthineers, Knoxville, TN, USA)	Prognostic value of automated vs. manual TMTV for PFS and OS	ICC = 0.68 (manual vs. auto TMTV); HR for PFS: 2.1 (auto), 3.3 (manual); Dice score = 0.65	Internal comparison of automated vs. manual segmentations; survival analysis with HRs and ICCs
Ritter (2022) [44]	Baseline PET/CT Radiomics	85 DLBCL patients (Center 1: training; Center 2: external test set)	Conventional PET metrics + radiomics (e.g., TLG, MTV, NGTDM busyness/coarseness, max diameter)	Automated machine learning (AutoML framework with feature selection and model training)	2-year event-free survival (EFS)	External test set: AUC = 0.85; Sensitivity = 79%; Specificity = 83%; NPV = 89%	External validation across 2 centers (Center 2 as test set); performance evaluated on EFS prediction
Santiago (2021) [38]	Contrast-enhanced CT Radiomics	52 DLBCL patients (26 refractory, 26 non-refractory); 180 lymph nodes; dual-reader segmentation	1218 handcrafted radiomic features from manually contoured lymph nodes (PyRadiomics); + nodal site + necrosis	Random Forest classifier with recursive feature elimination and 10-fold CV	Primary Treatment Failure (PTF)	AUC = 0.83 (Reader 1), 0.79 (Reader 2); Accuracy = 73%, Sensitivity = 62%, Specificity = 82%	70/30 train–test split; dual-reader reproducibility; model trained on Reader 1 and tested on Reader 2
Thiéry (2024) [25]	FDG-PET/CT + Clinical Data	545 patients from prospective multicenter cohort	Attributed lesion-graphs from baseline PET/CT (multiple lesion nodes); clinical tabular data (IPI, LDH, ECOG, etc.)	Graph Neural Network (GNN) with cross-attention fusion	2-year progression-free survival (PFS)	Outperformed clinical-only models (exact metrics not stated); interpretable lesion-level attribution	Internal validation; multiple attention configurations tested; interpretable model outputs
Yuan (2021) [42]	PET/CT Imaging	45 patients with DLBCL (multi-region scans: nasopharynx, chest, abdomen)	Multimodal image features from PET and CT, fused via hybrid learning module	Supervised CNN with hybrid feature fusion	Lesion segmentation in DLBCL	Mean DSC = 73.03%; MHD = 4.39 mm; superior to IL, MC, MB, QW baselines	Internal validation with ablation comparisons; region-specific accuracy (≥99%) reported
Zhao (2023) [37]	PET Radiomics + Clinical	240 patients from 2 centers (141 train, 61 internal test, 38 external test)	830 harmonized PET radiomics features from SUV4.0 segmentation + clinical data (selected via Pearson + LASSO)	Stacking ensemble (SVM, RF, GBDT, AdaBoost with RF meta-learner)	2-year PFS and OS	External test AUC: PFS = 0.771, OS = 0.725; Accuracy: PFS = 78.9%, OS = 76.3%	Internal and external validation; log-rank tests for KM stratification
Zhao (2025) [24]	PET Radiomics	175 elderly DLBCL patients (≥60 yrs), 1010 lesions, 2 centers	Baseline PET radiomics features, NCCN-IPI, BCL-2, TMTV	AutoML (AutoGluon), multivariable logistic regression	Treatment response	AUC (validation): Radscore = 0.712 vs. SUVmax = 0.616, MTV = 0.639, TLG = 0.657; combined model: significant (*p* < 0.05)	Train/validation cohorts; lesion-level + patient-level analysis
Zhou (2025) [39]	PET Radiomics + Clinical	522 DLBCL patients (response); 382 (2y-EFS); 1 center	Radiomics from 3 lesion selection + 5 segmentation methods; clinical data	XGBoost with RF feature selection	Treatment response; 2-year EFS	Combined model AUC = 0.908 (response), 0.837 (EFS); clinical only AUC = 0.622–0.636	Internal split; comparative AUC with Delong test

Legend: AI, artificial intelligence; ASPP, atrous spatial pyramid pooling; AUC, area under the curve; BCL-2, B-cell lymphoma 2; CE-T1WI, contrast-enhanced T1-weighted imaging; CHOP, cyclophosphamide, doxorubicin, vincristine, prednisone; CI, confidence interval; CNN, convolutional neural network; CNS, central nervous system; CT, computed tomography; CV, cross-validation; DCA, decision curve analysis; DLBCL, diffuse large B-cell lymphoma; DSC, Dice similarity coefficient; ECOG, Eastern Cooperative Oncology Group; EFS, event-free survival; ENI, extranodal involvement; FL, follicular lymphoma; FNN, fully connected neural network; GAINED, Groupe d’Etude des Lymphomes de l’Adulte trial; GBDT, gradient boosting decision tree; GNN, graph neural network; HR, hazard ratio; ICC, intraclass correlation coefficient; IL, input-level fusion; IPI, International Prognostic Index; KM, Kaplan–Meier; LASSO, least absolute shrinkage and selection operator; LDH, lactate dehydrogenase; MHD, modified Hausdorff distance; MIP, maximum-intensity projection; ML, machine learning; MB, multi-branch strategy; MC, multi-channel strategy; MTV, metabolic tumor volume; MVL, multi-view learning; NGTDM, neighbor gray tone difference matrix; NPV, negative predictive value; OS, overall survival; PARS, PET automated region segmentation; PCNSL, primary central nervous system lymphoma; PET, positron emission tomography; PH, proportional hazards; PFS, progression-free survival; QW, quantitative weighting; RF, random forest; SUV, standardized uptake value; SUVmax, maximum standardized uptake value; SVC, support vector classifier; SVM, support vector machine; TLG, total lesion glycolysis; TMTV, total metabolic tumor volume; VOI, volume of interest; WS, weak supervision; XGBoost, eXtreme gradient boosting.

**Table 7 medsci-13-00280-t007:** GEP innovative tools.

Tool Name (Author, Year)	Innovative Feature
RELB Anomaly Classifier (Carreras, 2024) [56]	A tool which identifies survival-linked anomalies via XGBoost and IHC validation
CAF Risk Model (Cui, 2025) [55]	A 13-gene CAF-based model with high 1–5y AUC prediction, validated across datasets
SurvIAE (Zaccaria, 2024) [57]	A combined autoencoder + MLP framework with explainability; MCC = 0.42 vs. R-IPI = 0.18
EcoTyper (Steen, 2021) [61]	A GEP + scRNA-seq integration to define TME ecosystems with prognostic relevance
MitoRG Risk Score (Wang, 2025) [59]	An 8-gene mitochondrial signature linked to OS and immune infiltration

Legend: AUC, area under the curve; CAF, cancer-associated fibroblast; GEP, gene expression profiling; IHC, immunohistochemistry; MCC, Matthews correlation coefficient; MitoRG, mitochondria-related gene; MLP, multilayer perceptron; OS, overall survival; R-IPI, Revised International Prognostic Index; scRNA-seq, single-cell RNA sequencing; TME, tumor microenvironment; XGBoost, extreme gradient boosting.

**Table 8 medsci-13-00280-t008:** GEP summary table.

First Author (Year)	Data Modality	Dataset Size/Source	Input Features	AI/ML Method	Outcome Predicted	Model Performance	Validation Strategy
Bentink (2008) [64]	Gene expression profiling	220 aggressive B-cell lymphomas (incl. 134 with OS data); validation on 303 external cases (GEO GSE4475)	Expression of 8 conserved oncogene-inducible modules (e.g., MYC.1, E2F3.1, RAS.4, SRC.2, etc.)	Semi-supervised clustering (ISIS), PAP derivation from binary module activation states	PAP subtype classification (BL-PAP, PAP-1 to PAP-4); Overall survival	PAP-1: HR = 0.25 (95% CI: 0.1–0.65); PAP-2: HR = 2.45 (95% CI: 1.16–5.17); E2F3.1 module: HR = 0.47 (*p* = 0.00003)	External validation on 303 patients using cross-platform transfer of PAP classifiers
Carreras (2024) [56]	Gene expression profiling	414 patients (GSE10846); external: TCGA, GSE57611, GSE31312, GSE117556	12 genes identified via anomaly detection (e.g., RELB, UBL7, HYAL2, IGFBP7, TRAPPC1) from apoptosis, MAPK, mTOR, and NF-κB pathways	Anomaly detection, ML classifiers (XGBoost, RF, ANN), Cox regression	Overall survival (OS)	XGBoost: 99.8% accuracy; RF: 98.6%; Cox: HYAL2/UBL7 = poor OS; TRAPPC1/IGFBP7/RELB = good OS (*p* < 0.01); RELB validated via GSEA and log-rank test	External validation in TCGA, GSE57611; anomaly signature tested in GSE31312, GSE117556; IHC in 30 DLBCL and 10 tonsils
Cui (2025) [55]	Gene expression profiling	412 patients (GSE10846 and GSE11318); validation: GSE53786	13 CAF-related genes from WGCNA module of 247 prognostic genes (e.g., FNDC1, IGFBP3, CSTA)	MCP-counter, ESTIMATE, WGCNA, LASSO Cox regression	Overall survival (OS)	AUC = 0.826 (1y), 0.808 (3y), 0.795 (5y); HR = 3.78 (training), HR = 4.01 (validation), both *p* < 0.001	Internal split (7:3 ratio), external validation in GSE53786; PCA, t-SNE, ROC, KM, nomogram, STROBE-compliant
Halder (2019) [67]	Gene expression profiling	77 samples (Harvard DLBCL dataset); 58 DLBCL, 19 FL	7129 gene expression features	ALRFC (Active Learning Rough-Fuzzy Classifier), compared with AL-MI, ALBT-MCSVM, FKNN, FRNN, etc.	Lymphoma subtype (DLBCL vs. FL)	ALRFC: Accuracy = 87.26%, Precision = 0.7992, Recall = 0.8478, F1 = 0.8170, Kappa = 0.5868; outperformed all baselines	Internal validation; model trained with 12–15 actively selected samples
He (2024) [60]	Gene expression (PCD-related)	5 GEO datasets (n = 207 total; 95 train, 72 test); cell line validation in VAL and IM-9	1074 DEGs from 1545 PCD-related genes	12 ML algorithms (XGBoost, GBM, etc.); 91 combinations	DLBCL molecular subtype (C1/C2), prognosis	AUC = 0.7–0.9; high F-scores in test sets	Internal train–test splits; external test cohorts; transcriptome sequencing validation in cell lines
Hopp (2015) [65]	Gene expression + DNA methylation	936 samples (MMML cohort): 5 lymphoma subtypes + B/GCB cells + cell lines	Methylation and expression profiles (Affymetrix U133A; centralized β-values); co-regulated gene clusters	Self-organizing maps (SOM); integrative multi-omics clustering	Lymphoma subtype-specific expression/methylation patterns	SOM modules captured subtype-specific regulatory modes (e.g., hyper/hypomethylation, expression–methylation correlations)	Internal profiling; cross-comparison between lymphoma subtypes and healthy B/GCB cell references
Merdan (2021) [58]	Gene expression (RNA-seq) + Immune infiltration	718 DLBCL patients (Reddy et al. cohort)	Normalized gene expression (≥12,000 genes), IPI, tumor-infiltrating immune cell fractions via CIBERSORT (no B-cells)	Hierarchical clustering, Lasso Cox regression, multivariable survival modeling	Overall survival (OS)	AUC = 0.78 (2y), 0.78 (5y), 0.80 (10y); HR = 2.45 for high- vs. low-risk groups; improved performance when combined with IPI	70:30 train/test split; multiple clustering strategies; validated with survival metrics, immune infiltration correlation, and KM/log-rank
Murphy (2022) [68]	Gene expression (microarray)	77 samples (58 DLBCL, 19 FL); 7129 genes	DEGs via limma; 250 top-ranked features	Enhanced BPSO (EBPSO) vs. standard BPSO + SVM + LOOCV	DLBCL vs. FL classification	Accuracy = 100% (EBPSO); signature = 5 genes; runtime = 684 s (vs. 1447 s for BPSO); smaller gene sets	10 repeated runs; compared EBPSO and BPSO on same system; evaluated stability, parsimony, and reproducibility
Qi (2022) [62]	Gene expression (microarray)	Training: 432 samples (5 GEO datasets); Testing: 420 (GSE10846) + ICB: GSE35640	CD8+ T cells, NK cells; 12 gene signatures (e.g., VGF, RAD54L)	RF, SVM, ANN with 5-fold CV; random search tuning	Immune subtype (IS vs. NIS); prognosis; ICB response	AUC = 0.948 (immune subtype classifier); IS had better OS and ICB response (57% vs. 19%)	External validation on GSE10846 (n = 420); immune function/ICB correlation; docking with ZINC15 compounds
Risueno (2020) [54]	Transcriptomics (FFPE)	414 GEP (GSE10846), 245 FFPE NanoString, 94 R/R in trial	Gene expression (26-gene classifier), TME immune composition	RFE-SVM, LDA, NTP, hierarchical clustering	Avadomide response, PFS, Immune-rich subgroup classification	Classifier-positive vs. -negative: ORR 44% vs. 19%, median PFS 6.2 vs. 1.6 months (HR = 0.49, *p* = 0.0096); Not predictive for R-CHOP or chemo	Internal training (nested CV), Affymetrix-to-NanoString replication, IHC confirmation, clinical validation (NCT01421524)
Shipp (2002) [5]	Gene expression profiling	77 samples (58 DLBCL, 19 FL); Affymetrix oligoarrays	6817 gene expression features from diagnostic biopsies	Supervised learning (Weighted Voting)	OS (cure vs. fatal/refractory); DLBCL vs. FL classification	5-year OS: 70% (predicted cured) vs. 12% (refractory); good separation of FL vs. DLBCL	Internal validation with outcome-based classifier development
Steen (2021) [61]	Transcriptomics + Single-cell RNA-seq	Bulk RNA-seq cohorts + scRNA-seq (size not specified)	Transcriptomic profiles of malignant B cells and 12 TME cell lineages	EcoTyper (ML framework combining deconvolution and clustering)	Identification of malignant B-cell and TME cell states and ecosystems	Stratified 5 malignant B-cell states and 12 TME lineages into ecosystems with prognostic relevance	Internal validation across multiple cohorts; integration with known COO/genotypic subtypes
Wang (2024) [69]	Transcriptomics (DRG-based profiling)	GSE31312 (n = 470), GSE12453 (DLBCL + normal B-cell subsets)	24 disulfidptosis-related genes (DRGs) → narrowed to 8 prognostic DRGs: CAPZB, DSTN, GYS1, IQGAP1, MYH9, NDUFA11, NDUFS1, OXSM	Unsupervised clustering, LASSO–Cox regression, risk score modeling, RF classifier	Prognostic stratification (OS), immune infiltration profiling	AUC = 0.716 (5-year OS), cluster 3 (low-risk) had best prognosis	Internal validation via consensus clustering, KM analysis, ROC, and immune landscape analysis
Wang (2025) [59]	Transcriptomics	GSE56315 (55 DLBCL, 33 normal); GSE10846 (training); GSE11318 and GSE87371 (validation)	1136 mitochondria-related genes (MitoRGs) → 305 DEGs → 8-gene prognostic signature (e.g., PCK2, NDUFA11, MYH9)	LASSO Cox regression, multivariate Cox, ssGSEA, ROC, Random Forest, consensus clustering	Overall survival (OS); immune infiltration; drug sensitivity	AUC = 0.79 (5-year OS, training); immune and drug response profiles stratified by risk group	External validation in GSE11318 and GSE87371; experimental validation (PCK2 knockdown)
Xu (2005) [66]	Transcriptomics (Microarray)	240 DLBCL patients (GSE10846/Rosenwald et al.): 160 training, 80 test	7399 gene expression features from diagnostic biopsy samples	PSO for feature selection + PNN classifier	Survival risk group (high vs. low)	80% accuracy (test); log-rank *p* < 0.0001 (train and test)	Training: LOOCV; Independent hold-out test set (n = 80)
Zaccaria (2024) [57]	Transcriptomics	GSE117556 (n = 928), GSE98588 (n = 137), Schmitz (n = 240), GSE181063 (n = 100+)	DEGs from 9737 shared genes (standardized), SHAP-ranked	Autoencoder (AE) + MLP (SurvIAE), SHAP for XAI	OS and PFS at 12, 36, 60 months	Best MCC = 0.42 (SurvIAE-S-PFS36) vs. 0.18 (R-IPI)	Multiset internal + external validation, comparison to R-IPI
Zhang (2025) [63]	Transcriptomics (M2 Macrophage-Associated Genes)	GEO (GSE9327, GSE23647, GSE32018, GSE83632; prognostic validation: GSE181063)	77 DEGs identified by CIBERSORT and WGCNA modules; 7 biomarkers selected: SMAD3, IL7R, IL18, FAS, CD5, CCR7, CSF1R	LASSO, SVM-RFE, Random Forest; Logistic Regression	Diagnosis and Prognosis of DLBCL	AUCs not explicitly reported; 5 genes (SMAD3, IL7R, IL18, FAS, CD5) significantly predicted OS (*p* < 0.01); HRs 0.78–0.87	Prognostic value assessed using Kaplan–Meier and univariate Cox regression in GSE181063
Zhao (2016) [70]	Transcriptomics/Gene Expression	414 (training) + 855 (validation); GEO datasets	Expression of 8 genes (MYBL1, LMO2, BCL6, MME, IRF4, NFKBIZ, PDE4B, SLA)	SVM with ROC-derived cutoffs	Cell-of-origin subtype (GCB vs. non-GCB)	Concordance with GEP: 94.0% (train), 91.0%/94.4% (validation); significant OS difference by subtype	External validation in 2 independent GEO cohorts (total n = 855); multivariate analysis for IPI independence
Zhuang (2024) [71]	Transcriptomics (Gene Expression)	Training: 414 (GSE10846); Validation: GSE34171, GSE87371, GSE31312	Expression of 7 genes (SERPING1, TIMP2, NME1, DCTPP1, RFC4, POLE2, SNRPD1); subtype (IME vs. CCE)	LASSO, Random Forest, Point-Biserial Correlation for feature selection; SVM for classification	Molecular subtype and OS prognosis	AUC = 0.973, Accuracy = 88.6% (GSE10846); HR > 1.4, *p* < 0.05 in validation datasets	External validation in 3 independent datasets (GSE34171, GSE87371, GSE31312); multivariate Cox regression applied

Legend: AE, autoencoder; AUC, area under the curve; ANN, artificial neural network; ALRFC, active learning rough-fuzzy classifier; BPSO, binary particle swarm optimization; CCE, cell-cycle-enriched; CI, confidence interval; COO, cell of origin; CSF1R, colony-stimulating factor 1 receptor; DLBCL, diffuse large B-cell lymphoma; DEG, differentially expressed gene; DRG, disulfidptosis-related gene; E2F3.1, E2F transcription factor 3 module 1; EBPSO, enhanced binary particle swarm optimization; FFPE, formalin-fixed paraffin-embedded; FL, follicular lymphoma; F1, F1-score; GCB, germinal center B-cell like; GEP, gene expression profiling; GEO, Gene Expression Omnibus; HR, hazard ratio; IHC, immunohistochemistry; IME, immune-enriched; IPI, International Prognostic Index; IRF4, interferon regulatory factor 4; IS, immune subtype; LASSO, least absolute shrinkage and selection operator; LOOCV, leave-one-out cross-validation; MCC, Matthews correlation coefficient; ML, machine learning; MLP, multi-layer perceptron; MMML, Molecular Mechanisms in Malignant Lymphoma; MYBL1, MYB proto-oncogene like 1; NanoString, NanoString gene expression platform; NIS, non-immune subtype; NTP, nearest template prediction; OS, overall survival; PAP, prognostically associated profile; PCD, programmed cell death; PDE4B, phosphodiesterase 4B; PFS, progression-free survival; PNN, probabilistic neural network; PSO, particle swarm optimization; RF, random forest; RFE, recursive feature elimination; ROC, receiver operating characteristic; R-IPI, Revised International Prognostic Index; RNA-seq, RNA sequencing; scRNA-seq, single-cell RNA sequencing; SHAP, SHapley Additive exPlanations; SLA, Src-like adaptor; SNRPD1, small nuclear ribonucleoprotein D1; SOM, self-organizing map; SRC, SRC proto-oncogene module; ssGSEA, single-sample gene set enrichment analysis; STROBE, Strengthening the Reporting of Observational Studies in Epidemiology; SVM, support vector machine; TME, tumor microenvironment; UC, unclassified; WGCNA, weighted gene co-expression network analysis; XAI, explainable artificial intelligence; XGB, extreme gradient boosting; ZINC15, compound screening database.

**Table 9 medsci-13-00280-t009:** Innovative approaches to specific prognostic genetic mutations in DLBCL.

Tool Name (Author, Year)	Novel Feature
Modified Naïve Bayes (Albitar, 2022) [76]	A layered RNA-seq-based 4-group OS classifier, validated across 626 patients
10-algorithm ML framework (Du, 2024) [72]	An assessment tool for TP53 mutation-aware risk model; it integrates VAF, structure, and mutation class
Lactylation RiskScore (Zhu, 2024) [73]	A prognostic model integrating lactylation genes and immune markers of the tumoral microenvironment
Mitochondria RiskScore (Zhou, 2025) [74]	An 18-gene signature linked to PD-L1, CD20, and therapy response
Multi-omics clustering (Liang, 2025) [75]	A tool documenting ODC1 expressors as an aggressive subgroup with immune suppression

Legend: RNA-seq, RNA sequencing; OS, overall survival; ML, machine learning; TP53, tumor protein p53; VAF, variant allele frequency; PD-L1, programmed death-ligand 1; CD20, cluster of differentiation 20; ODC1, ornithine decarboxylase 1.

**Table 10 medsci-13-00280-t010:** Specific genetic mutations—summary table.

First Author (Year)	Data Modality	Dataset Size/Source	Input Features	AI/ML Method	Outcome Predicted	Model Performance	Validation Strategy
Albitar (2022) [76]	RNA expression + mutations	626 DLBCL patients (379 nodal, 247 extranodal)	Expression of 1408 genes (RNA-seq from FFPE)	Modified Naïve Bayes + 12-step CV	Survival group classification (4 levels)	HR = 0.237 (2-group model), HR = 0.174 (4-group model); validation: HR = 0.26 (2-group), HR = 0.53 (4-group), all *p* < 0.01	Internal training on 379 nodal patients; external validation on 247 extranodal DLBCL; final test on 1/3 of pooled 626-patient set
Carreras (2021) [77]	Gene Expression + IHC	414 patients (GSE10846) + independent Tokai cohort	54,613 probes reduced to 16-gene signature + PD-L1, IKAROS, BCL2, MYC, CD163, TNFAIP8	MLP, RBF ANN, logistic regression, Bayesian network, decision trees (CHAID, C&R, QUEST), GSEA	Overall survival, progression-free survival	Final model accuracy: 82.1%; high PD-L1 = poor OS/PFS; high IKAROS = good OS/PFS	Independent validation via IHC quantification and survival stratification
Carreras (2021b) [78]	IHC + Gene Expression	97 DLBCL cases (Tokai Univ); validation: 414 LLMPP cases	Caspase-8 IHC (active p18), related markers (cCASP3, cPARP, E2F1, TP53, TNFAIP8, BCL2, MDM2, etc.)	CHAID tree, Bayesian network, discriminant analysis, C5 tree, logistic regression, MLP, RBF NN	OS, PFS, Caspase-8 expression modeling	>80% accuracy across multiple ML methods	Independent gene expression validation in LLMPP (n = 414); white-box explainability emphasized
Carreras (2021c) [79]	Gene Expression (nCounter)	106 DLBCL cases; external validation on 414 GSE10846	730 immune-oncology genes	Multilayer perceptron, RBF neural network, SVM, LR, C5, CHAID, KNN, Bayesian network	OS, COO (GCB vs. ABC)	AUC = 0.98 (OS); AUC = 1 (subtype)	External validation on GSE10846 (414 cases); IHC correlation; multivariate analysis
Carreras (2021d) [80]	IHC + Transcriptomics	198 DLBCL cases (Tokai Hospital) + GSE10846	CSF1R expression patterns (TAMs vs. diffuse), 10 CSF1R-related markers, transcriptomic correlates	Multilayer perceptron, SVM, regression models	CSF1R pattern prediction; PFS correlation	ML models showed high accuracy; CSF1R-TAMs associated with poor PFS	Internal validation (Tokai cohort); GEO dataset correlation
Dai (2024) [87]	Transcriptomics + Functional Genomics	47 DLBCL patients (TCGA); CRISPR in Raji and SLVL cell lines	COPS5 co-expression and survival; CRISPR knockout	Correlation analysis, survival modeling	Proliferation, overall survival	High COPS5 linked to poor OS (*p* = 0.0168); growth inhibition in KO models	Kaplan–Meier on TCGA; CRISPR-Cas9 in vitro validation
de Groen (2025) [88]	Genomics + Transcriptomics	106 patients (PB-DLBCL n = 52; polyostotic-DLBCL n = 20; nodal GCB-DLBCL n = 34)	GEP, genomic aberrations, immune gene sets, TME composition	Unsupervised clustering, ssGSEA, CIBERSORTx	Immune TME subtype, survival	Immune-rich cluster associated with superior survival (*p* = 0.030); *p* < 0.001 for immune profiling	Transcriptomic profiles validated via IHC (CD3, FOXP3); immune subtypes correlated with clinical outcome
Du (2024) [72]	Genomic + Transcriptomic	2637 public DLBCL pts + 108 JSPH pts; 21 CCLE cell lines	TP53 mutation types, functional class, CNV, VAF, RNA expression	10 ML methods: LASSO, Ridge, CoxBoost, RSF, Enet, survival-SVM, GBM, plsRcox, SuperPC, stepwise Cox (150 combos)	PFS, OS stratification in TP53-mutated patients	Best models selected by C-index; high VAF, non-missense, LOF/DNE and multi-site mutations associated with poor survival	5 public + 1 internal + 1 external JSPH cohort; 4:1 train-test split, 10-fold CV
Liang (2025) [75]	Multi-omics (genome, transcriptome, scRNA-seq)	2133 patients (public datasets)	ODC1 expression, CNVs (8p23.1, 9p21.3), stemness/TME markers	ML-based clustering, risk stratification	High-risk subgroup identification, OS, PFS	3y OS = 54.3% vs. 83.6%, *p* < 0.0001	Internal validation, in vitro and single-cell experiments
Loeffler-Wirth (2019) [84]	Transcriptomic profiling	873 B-cell lymphomas (MMML consortium)	Microarray gene expression data	Self-Organizing Map (SOM) clustering	Molecular subtyping; prognostic stratification	Modular PAT classification; phenotypic similarity trees	Unsupervised learning with clinical/pathologic correlation
Orgueira (2020) [82]	Transcriptomics + Clinical	233 training (GSE10846), 64 test (GSE23501)	4-gene cluster (TNFRSF9, BIRC3, BCL2L1, G3BP2), 50-gene RF signature, COO, clinical variables	Random Forest, Mclust clustering	Overall Survival	c-index = 0.84 (train), 0.79 (test)	External validation on independent GEO cohort
Peng (2024) [81]	Clinical + Molecular	401 patients (single center)	22 variables including MYC, LDH, AMC, PLT, extranodal sites	Random Survival Forest + Bi-LSTM + Logistic Hazard	OS, PFS	McPM: high C-index, low IBS; sMcPM outperformed IPI for PFS (*p* < 0.0001 vs. 0.44)	Internal validation (train/test split); model comparison with IPI
Stokes (2024) [83]	Transcriptomics + CNAs + Immune scores	1208 training (MER, REMoDL-B, GOYA, Reddy replication cohorts)	RNA-seq expression, MSigDB pathways, immune cell type scores, CNAs	iClusterPlus (unsupervised clustering); SubLymE (multinomial GLM classifier)	Molecular subtyping and EFS/OS	A7 cluster had significantly inferior EFS (e.g., MER HR = 2.00, *p* = 0.006); validated across 4 cohorts	External validation in MER, REMoDL-B, GOYA, Reddy cohorts; multivariate Cox models
Tyryshkin (2023) [85]	Transcriptomics	Training: 121 (FFPE, KHSC); Validation: 569 (EGA RNA-seq)	23 transcripts (TCF3-regulated + others)	k-Nearest Neighbors (kNN), clustering	Overall Survival	HR 2.29 (Group A vs. B), C-index 0.70	External validation on 569 patients from public RNA-seq data
Xu-Monette (2020) [86]	Targeted RNA-Seq + clinical	418 DLBCL patients	Transcriptomic + pathogenetic features	Deep learning (unspecified)	COO subtype, OS prediction	High concordance with Affymetrix and Lymph2Cx COO; NGS survival model stratified 30% as high-risk with poor survival	External validation in 2 independent cohorts
Zhang (2020) [4]	Targeted Genomics (NGS)	342 DLBCL patients, single-center cohort	Mutations/translocations in 46 genes (e.g., MYC, BCL2, BCL6, MYD88, CD79B)	Random Forest	Molecular subgroups; OS stratification	MYC-trans signature: independent adverse factor; worse OS in 3+ sig. group	Internal comparison with Schmitz classification; multivariate survival analysis
Zhou (2025) [74]	Transcriptomics—Mitochondria-Related Genes	GSE10846 (n = 412, training); GSE11318 (n = 199); GSE53786 (n = 119)	1136 MRGs from MitoCarta3.0; 18 prognostic genes (e.g., DNM1L, COX7A1, PDK3, CD20, PD-L1); clinical features (age, stage, LDH, ECOG)	Lasso–Cox regression; k-means clustering; nomogram construction; ROC analysis	OS; immune microenvironment; therapy response prediction	AUC (OS): GSE10846—0.787 (1y), 0.809 (3y), 0.792 (5y); GSE11318—0.715/0.754/0.768; GSE53786—0.815/0.781/0.724; Nomogram AUCs > 0.81 across all sets	External validation (GSE11318, GSE53786); KM analysis; multivariate Cox regression; time-dependent ROC
Zhu (2024) [73]	Transcriptomics (Lactylation genes)	TCGA (n = 47), GSE87371 (n = 221), GSE32918 (n = 244)	Expression of lactylation-related genes; clinical data (age, stage, ECOG, LDH); immune infiltration; CD20, PD-L1 expression	LASSO + Cox regression; RiskScore model	OS stratification	AUC (1y/3y/5y): 0.787/0.809/0.792 (train); 0.715–0.840 (val)	External validation in 2 GEO cohorts + TCGA

Legend: AUC, area under the curve; ANN, artificial neural network; AMC, absolute monocyte count; BCL2, B-cell lymphoma 2; BCL6, B-cell lymphoma 6; Bi-LSTM, bidirectional long short-term memory; CD, cluster of differentiation; CD79B, B-cell antigen receptor complex-associated protein beta chain; CD163, hemoglobin scavenger receptor; CHAID, Chi-squared Automatic Interaction Detector; CNAs, copy number alterations; CNV, copy number variation; COO, cell of origin; CRISPR, clustered regularly interspaced short palindromic repeats; CSF1R, colony-stimulating factor 1 receptor; CV, cross-validation; DLBCL, diffuse large B-cell lymphoma; DNE, dominant-negative effect; DNM1L, dynamin-1-like protein; ECOG, Eastern Cooperative Oncology Group; EFS, event-free survival; FFPE, formalin-fixed paraffin-embedded; FOXP3, forkhead box P3; GCB, germinal center B-cell; GEP, gene expression profiling; GLM, generalized linear model; GOYA, a phase III clinical trial cohort; GSEA, gene set enrichment analysis; HR, hazard ratio; IHC, immunohistochemistry; IKAROS, zinc finger transcription factor IKZF1; IPI, International Prognostic Index; LASSO, least absolute shrinkage and selection operator; LLMPP, Lymphoma/Leukemia Molecular Profiling Project; LOF, loss of function; MDM2, mouse double minute 2 homolog; MER, Molecular Epidemiology Resource cohort; MMML, Molecular Mechanisms in Malignant Lymphoma consortium; ML, machine learning; MLP, multilayer perceptron; MYC, MYC proto-oncogene; MYD88, myeloid differentiation primary response 88; NGS, next-generation sequencing; OS, overall survival; PAT, phenotypic activation tree; PDK3, pyruvate dehydrogenase kinase isoform 3; PD-L1, programmed death-ligand 1; PFS, progression-free survival; PLT, platelet count; plsRcox, partial least squares regression for Cox model; p18, active form of caspase-8; RF, random forest; RBF, radial basis function; ROC, receiver operating characteristic; RNA, ribonucleic acid; RNA-seq, RNA sequencing; RSF, random survival forest; scRNA-seq, single-cell RNA sequencing; SVM, support vector machine; TCGA, The Cancer Genome Atlas; TCF3, transcription factor 3; TME, tumor microenvironment; TP53, tumor protein p53; TNFAIP8, tumor necrosis factor alpha-induced protein 8; VAF, variant allele frequency.

**Table 11 medsci-13-00280-t011:** microRNA profiling in DLBCL—summary table.

First Author (Year)	Data Modality	Dataset Size/Source	Input Features	AI/ML Method	Outcome Predicted	Model Performance	Validation Strategy
Minezaki (2020) [89]	microRNA profiling (vitreous/serum)	14 VRL vs. 78 controls (uveitis, macular hole, ERM, healthy)	17 differentially expressed miRNAs	Random Forest	VRL diagnosis vs. controls	Best AUC = 0.921 (miR-361-3p), Accuracy = 0.875	Internal cross-validation
Nakamura (2023) [90]	Circulating miRNA	152 DLBCL (128 responders/24 non-responders from GSE21848 and GSE40239)	448 miRNAs analyzed; 36-miRNA panel via Boruta	11 classifiers incl. RF, SVM, GBDT, LR, NB	R-CHOP response classification	Best AUC = 0.751 (Boruta + RF, 36 miRNAs)	Double cross-validation

Legend: AUC, area under the curve; DLBCL, diffuse large B-cell lymphoma; ERM, epiretinal membrane; GBDT, gradient boosting decision tree; LR, logistic regression; miRNA, microRNA; NB, Naïve Bayes; RF, random forest; R-CHOP, rituximab, cyclophosphamide, doxorubicin, vincristine, prednisone; SVM, support vector machine; VRL, vitreoretinal lymphoma.

**Table 12 medsci-13-00280-t012:** Circulating tumor DNA (ctDNA)—summary table.

First Author (Year)	Data Modality	Dataset Size/Source	Input Features	AI/ML Method	Outcome Predicted	Model Performance	Validation Strategy
Meriranta (2022) [91]	ctDNA	101 patients with high-risk DLBCL (FINNISH trial)	ctDNA burden, mutation profile, fragmentation features	Regularized logistic regression	Survival (OS, PFS), MRD monitoring	AUROC 0.86 (2-year OS prediction)	Internal validation (training/test split)
Mutter (2022) [92]	ctDNA	92 CNSL + 44 non-CNSL + 24 healthy controls (160 total)	ctDNA from CSF, plasma, tumor samples	Mutation profiling + ML (CAPP-Seq-based)	CNSL vs. non-CNSL classification	Sensitivity: 59% (CSF), 25% (plasma); High PPV	Internal validation
Zhao (2024) [93]	ctDNA (IgH VDJ rearrangement)	55 DLBCL patients with ctDNA and/or tissue data from China	Dominant circulating/tissue-matched clonotype %, clinical features (e.g., extranodal involvement, IPI, LDH)	Decision Tree	Progression after R-CHOP	AUC = 0.85 (tissue-matched); sensitivity = 0.85; specificity = 0.78	Internal (10-fold cross-validation)

Legend: AUC, area under the curve; AUROC, area under the receiver operating characteristic curve; CNSL, central nervous system lymphoma; CSF, cerebrospinal fluid; ctDNA, circulating tumor DNA; DLBCL, diffuse large B-cell lymphoma; IgH, immunoglobulin heavy chain; IPI, International Prognostic Index; LDH, lactate dehydrogenase; ML, machine learning; MRD, minimal residual disease; OS, overall survival; PFS, progression-free survival; PPV, positive predictive value; R-CHOP, rituximab, cyclophosphamide, doxorubicin, vincristine, prednisone.

**Table 13 medsci-13-00280-t013:** Multi-omics—summary table.

First Author (Year)	Data Modality	Dataset Size/Source	Input Features	AI/ML Method	Outcome Predicted	Model Performance	Validation Strategy
Futschik (2003) [94]	Multi-omics (Gene expression + clinical)	58 DLBCL patients (Affymetrix microarray + IPI)	Gene expression (Affymetrix), IPI score	EFuNN, SVM, kNN, weighted voting, Bayesian classifier	Treatment outcome/survival	Accuracy: 70.7–87.5%	Internal validation
Mosquera Orgueira (2022) [95]	Transcriptomics + Mutations + Clinical	481 patients/UK HMRN cohort (GSE181063)	17 gene expression features (LymForest-25), IPI, COO, MHG, mutational clusters	PCA, Random Forest, Cox regression	Overall survival (OS)	Best AUCs: 0.82 (5y), 0.81 (0.5–2y) in ≤70y subgroup (LymForest + IPI + COO + MHG)	Bootstrapped internal validation (500 cycles); subgroup analysis by age
Mosquera Orgueira (2023) [96]	Transcriptomics + Clinical	REMoDL-B trial (928 patients: 469 R-CHOP, 459 RB-CHOP)	17-gene expression (modified LymForest-25), CD3 markers, clinical features	Random Forest	Progression-Free Survival (PFS)	C-index: 0.668 (R-CHOP), 0.631 (RB-CHOP); HR 0.70 in 50% high-risk (*p* = 0.03)	Internal validation; post hoc subgroup analysis
Goedhart (2025) [97]	CNV + Mutations + Clinical	101 DLBCL patients (uniformly treated)	67 CNVs, 69 mutations, 3 translocations, IPI	EB-coBART (Bayesian trees)	2-year PFS (binary)	C-index = 0.714	Internal validation (held-out split)
Ouyang (2025) [98]	Transcriptomics (lncRNA)	831 (TCGA, GSE10846, GSE11318, GSE23501, GSE53786)	Cuproptosis-related lncRNAs (n = 126), selected by RF, LASSO, Boruta	Transformer-based model + Bagging Ensemble	Overall survival	Internal AUC: 0.79 (1y), 0.83 (3y), 0.82 (5y); External AUC: 0.66–0.69	5-fold cross-validation, external validation

Legend: AUC, area under the curve; CNV, copy number variation; COO, cell of origin; DLBCL, diffuse large B-cell lymphoma; EB-coBART, empirical Bayes conditional Bayesian additive regression trees; EFuNN, evolving fuzzy neural network; GSE, Gene Expression Omnibus Series; HR, hazard ratio; HMRN, Haematological Malignancy Research Network; IPI, International Prognostic Index; lncRNA, long non-coding RNA; MHG, molecular high grade; OS, overall survival; PCA, principal component analysis; PFS, progression-free survival; RF, random forest; RB-CHOP, rituximab, bendamustine, cyclophosphamide, doxorubicin, vincristine, prednisone; REMoDL-B, R-CHOP vs. RB-CHOP trial; R-CHOP, rituximab, cyclophosphamide, doxorubicin, vincristine, prednisone; SVM, support vector machine; TCGA, The Cancer Genome Atlas.

**Table 14 medsci-13-00280-t014:** Frequently reported AI/ML-derived prognostic parameters in DLBCL across modalities.

No.	Parameter/Feature	Data Type	Representative Identified Source Studies	Endpoint(s)	Observed Particular Features
1	Age	Clinical	Biccler et al. (Nordic registries) [8]; Shen et al. [13]; Zhao et al. (SEER) [14]	OS, long-term mortality	Consistently retained in RSF, stacked Cox, and LASSO; baseline host factor that improved C-index from ~0.70 (IPI) to ~0.74–0.76 when combined with other vars.
2	LDH (serum)	Clinical	Shen et al. [13]; Zhu et al. (R/R cohort) [15]; Czibor et al. (PET + labs) [33]	OS, PFS	Proxy for tumor burden; repeatedly ranked among top features even in imaging-augmented models; present in high-AUC models (0.83 for 24-mo PFS in [33]).
3	ECOG performance status	Clinical	Biccler et al. [8]; Shen et al. [13]	OS	One of only two variables (age + ECOG) that could rival IPI in Biccler; included in stacked survival models with C-index 0.744–0.756.
4	Extranodal involvement/disease spread	Clinical	Zhao et al. (composite lymphomas) [14]; PET dissemination work (Girum et al.) [28,48]	OS, PFS, treatment failure	Clinical counterpart of radiomic dissemination; AI models tended to give weight to “spread” in both structured and image-derived form.
5	Baseline PET total metabolic tumor volume (TMTV)/surrogate TMTV	PET/CT radiomics	Capobianco et al. [23]; Girum et al. [48]; Ferrández et al. [27,43]; Jiang et al. [51]	PFS, OS	Core radiomic burden marker; LASSO–Cox models around TMTV reached C-index ~0.74 (OS) and ~0.72 (PFS) [23]; CNN/MIP approaches improved over IPI (AUC 0.66 vs. 0.60) [43].
6	Lesion dissemination metrics (Dmax, spleen-referenced distance, SpreadSpleen)	PET/CT radiomics (spatial)	Girum et al. (REMARC, spleen-relative features) [28]; Girum et al. (MIP + AI) [48]; Zhao et al. (stacking) [37]	PFS, early events	Adding spleen-referenced spread improved model C-index over TMTV alone by ~0.05–0.07 and identified very high-risk groups (HR~11–12) [28,48].
7	MTVrate/volumetric texture composite	PET/CT radiomics	Czibor et al. [33]; Chen et al. (DFR + AutoML) [26]; Jiang et al. [51]	24-mo PFS, OS	The MTVrate alone gave AUC 0.74 [33], and the ML model combining it with clinical data reached AUC 0.83; deep-feature radiomics had C-index 0.739–0.784 for PFS [26].
8	Multigene/transcriptomic risk signatures (CAF-related, SurvIAE latent features, immune-TME signatures)	Transcriptomics/multi-omics	Cui et al. (CAF 13-gene) [55]; Zaccaria et al. (SurvIAE) [57]; Merdan et al. (immune infiltration) [58]; Wang (mitochondrial) [59]; Zhou (mitochondria-related signature) [74]	OS (1–5 years)	These models frequently reported C-index/AUC in the 0.78–0.87 range, clearly above IPI (~0.60–0.70); they capture biology not present in clinical data.
9	TP53 mutation-aware composite (mutation class, VAF, multi-site)	Genomic	Du et al. [72]	OS, PFS (TP53-mutated subset)	10-method ML pipeline showed that TP53 features separated truly high-risk cases within TP53-mutated DLBCL; best models selected by C-index added granularity to “genetic high risk.”
10	ctDNA burden/fragmentation profile	Liquid biopsy	Meriranta et al. [91]	OS, MRD-related outcomes	Regularized models using ~60 ctDNA-derived variables and reached AUROC 0.86 for 2-year OS—competitive with imaging/multi-omics.
11	MYC/FISH surrogates from WSI or IHC-derived COO	Digital/conventional pathology	Swiderska-Chadaj et al. (MYC from H&E) [17]; Da Costa et al. (IHC-based COO) [20]	Risk groups, OS	AI replaced expensive tests in triage; sensitivity 0.90–0.95 for MYC-positive cases [17]; IHC-fed decision tree had κ = 0.83 vs. GEP and was prognostic [20].
12	Combined PET/CT deep features + NCCN-IPI	Multimodal	Chen et al. (deep-feature PET/CT + AutoML) [26]	OS, PFS	Fusion model: C-index 0.784 (PFS) and 0.831 (OS) train cohort, 0.739/0.782 validation cohort—better than NCCN-IPI alone (≈0.70–0.72).
13	Time to relapse/R/R setting variables (CD5+, albumin, β2-MG)	Clinical + PET (R/R)	Zhu et al. (R-ICE + ibrutinib) [15]	OS, PFS (1–3 years)	Gradient boosting and Cox-XGBoost reached OS AUC 0.863–0.898 and PFS AUC ~0.77–0.78; CD5+, LDH, albumin, β2-MG repeatedly selected.

Legend: AUC, area under the curve; β2-MG, beta-2 microglobulin; CAF, cancer-associated fibroblast; CNN, convolutional neural network; COO, cell of origin; CT, computed tomography; ctDNA, circulating tumor DNA; DLBCL, diffuse large B-cell lymphoma; Dmax, maximum dissemination distance; ECOG, Eastern Cooperative Oncology Group; FISH, fluorescence in situ hybridization; HR, hazard ratio; IPI, International Prognostic Index; LDH, lactate dehydrogenase; LASSO, least absolute shrinkage and selection operator; ML, machine learning; NCCN-IPI, National Comprehensive Cancer Network International Prognostic Index; OS, overall survival; PFS, progression-free survival; PET/CT, positron emission tomography/computed tomography; TMTV, total metabolic tumor volume; TP53, tumor protein p53; WSI, whole-slide image; XGBoost, extreme gradient boosting.

## Data Availability

No new data were created or analyzed in this study.

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
