# Peer review of "Artificial Intelligence for Risk Stratification in Diffuse Large B-Cell Lymphoma: A Systematic Review of Classification Models and Predictive Performances"

_medsci, 2025, doi:10.3390/medsci13040280_

Round 1

Reviewer 1 Report

Comments and Suggestions for Authors

Congratulations to the authors for the idea of ​​creating this systematic review, very interesting and useful for researchers, practitioners, and medical device and reagent companies!

Recommendations:

  1. The authors should also mention bispecific antibodies in the Introduction, among the types of curative salvage therapy, along with stem cell transplantation and chimeric antigen receptor T-cell (CAR-T) therapy.
  2. Information existing in the text should not be repeated in tables. Therefore, the text or tables should be carefully restructured.
  3. There is a repetition of terms in line 430: ”a decision-tree model that used used CD10”. Authors must correct minor grammatical errors throughout the article.
  4. References are not written homogeneously. For example: either 3370–81 or 3370-3381.

Author Response

Dear Editor-in-Chief,
Dear Academic Editor and Guest Editors,
Dear Reviewer,

We sincerely thank you for taking the time to review our manuscript entitled “Leveraging AI for Precision Prognostics in DLBCL: A Scoping Systematic Review of Emerging Approaches.” We are grateful for your thoughtful comments and constructive suggestions, which have helped us further improve the clarity, accuracy, and overall quality of our paper.

Below, we provide a detailed, point-by-point response addressing each of your observations. All modifications made to the manuscript are highlighted in yellow for ease of reference.

  1. The authors should also mention bispecific antibodies in the Introduction, among the types of curative salvage therapy, along with stem cell transplantation and chimeric antigen receptor T-cell (CAR-T) therapy.

Response: Thank you for your valuable suggestion. As advised, we have added a paragraph in the Introduction summarising the role of bispecific antibodies among curative salvage therapies, alongside autologous/allogeneic stem cell transplantation and chimeric antigen receptor T-cell (CAR-T) therapy. This addition highlights recent therapeutic advances in relapsed/refractory DLBCL and provides a more comprehensive overview of the evolving treatment landscape. The new text is highlighted in yellow in the revised manuscript.

      2. Information existing in the text should not be repeated in tables. Therefore, the text or tables should be carefully restructured.

Response: Thank you for this thoughtful suggestion. We fully understand the concern regarding redundancy between the text and tables. However, we would like to point out that, despite our efforts to minimise repetition, the online compact format of the journal allows for a reading experience in which tables can be accessed as complementary, pop-up summaries to the main text. Given the high density and technical complexity of the information, we believe that presenting key data both in text and in structured tables enhances clarity and supports readers who seek a concise overview without navigating lengthy narrative sections. Therefore, although we acknowledge that the tables add to the overall volume of the paper, they are intentionally designed to complement the written content and improve accessibility and comprehension for the reader.

    3. There is a repetition of terms in line 430: ”a decision-tree model that used used CD10”. Authors must correct minor grammatical errors throughout the article.

Response: Thank you for your observation. The typographical error in line 430 (“used used CD10”) has been corrected as suggested. In addition, the entire manuscript has been carefully rechecked multiple times to identify and correct any remaining spelling, grammatical, or readability issues. We are confident that the current version now reflects a clear and polished language standard suitable for publication.

      4. References are not written homogeneously. For example: either 3370–81 or 3370-3381.

Response: Thank you for your observation. All references have been reformatted to ensure consistency and compliance with the MDPI reference style guidelines. The reference list has been regenerated and standardised using RIS management software, ensuring uniform formatting (e.g., page ranges, punctuation, and citation style) throughout the manuscript.

We sincerely thank you for the constructive feedback and valuable observations, which have greatly contributed to improving the clarity, coherence, and overall quality of our manuscript.

Reviewer 2 Report

Comments and Suggestions for Authors

Overall Impression: This is a fantastic and highly relevant scoping review addressing a critical gap in DLBCL prognostication. The concept of leveraging artificial intelligence (AI), machine learning (ML), and deep learning (DL) to manage the known biological and clinical heterogeneity of DLBCL is a novel and compelling approach. The manuscript is well-structured and interesting to read, successfully demonstrating the diverse methods currently being explored in this rapidly evolving field. I commend the authors for undertaking this systematic effort.

Major Comment - Focus on Actionable Parameters:

While the review successfully maps the landscape of AI in DLBCL, its true clinical utility and the highest value of this research would be realized through a more granular and prescriptive focus on the specific input parameters validated across the literature. I was expecting a more descriptive and synthesized summary of the exact features (parameters) that the most successful DL/ML models use to predict patient outcome (e.g., progression-free survival (PFS) or overall survival (OS)).

The discussion should transition from which types of data (e.g., clinical, genomic, radiomic) are used to what exact features within those datasets are consistently identified as the strongest predictors by the AI models.

To maximize the impact, I recommend the authors create an explicit, structured table that catalogs the top 10-15 most frequently or most successfully integrated prognostic parameters (e.g., specific genomic mutations, radiomic features, or refined clinical metrics) identified by the reviewed AI/ML studies. This would transform the review into an actionable resource for subsequent clinical trial design and model development.

Minor Comments:

IPI Comparison: The discussion adequately highlights the limitations of the IPI, but a clearer synthesis of how much AI models outperform IPI/R-IPI across various studies would strengthen the argument for adoption.

Model Transparency: A brief discussion or a section dedicated to the explainability (XAI) of the reviewed models would be beneficial, as clinical adoption requires trust and understanding of the AI's decision-making process.

Conclusion: This manuscript is an excellent contribution to the field of hemato-oncology and computational medicine. Addressing the need for greater specificity regarding the actionable input parameters would significantly enhance the manuscript’s value and fulfill the promise of precision prognostics. The work is ready for publication pending a satisfactory revision focusing on parameter synthesis.

Author Response

Dear Editor-in-Chief,
Dear Academic Editor and Guest Editors,
Dear Reviewer,

We are deeply grateful for your positive and encouraging assessment of our manuscript entitled “Leveraging AI for Precision Prognostics in DLBCL: A Scoping Systematic Review of Emerging Approaches.” Your insightful feedback has been invaluable in strengthening the clarity, depth, and clinical applicability of our work. We sincerely appreciate your recognition of the review’s structure, relevance, and contribution to the field of computational hemato-oncology.

Below, we provide a detailed, point-by-point response to your observations. All corresponding modifications in the revised manuscript are highlighted in yellow.

Major Comment - Focus on Actionable Parameters:

While the review successfully maps the landscape of AI in DLBCL, its true clinical utility and the highest value of this research would be realised through a more granular and prescriptive focus on the specific input parameters validated across the literature. I was expecting a more descriptive and synthesized summary of the exact features (parameters) that the most successful DL/ML models use to predict patient outcome (e.g., progression-free survival (PFS) or overall survival (OS)).

The discussion should transition from which types of data (e.g., clinical, genomic, radiomic) are used to what exact features within those datasets are consistently identified as the strongest predictors by the AI models.

Response: Thank you for this excellent and constructive suggestion. As advised, the Discussion section has been updated and now includes a dedicated subsection entitled “Synthesis of Findings and Domain-Specific Trends: Superiority of AI/ML over Traditional Prognostic Indices.” This subsection provides a granular synthesis of the most promising parameters repeatedly validated across the reviewed studies and highlights their comparative performance against conventional benchmarks, such as the International Prognostic Index (IPI) and revised IPI (R-IPI).

Within this new section, we emphasise the most relevant input features associated with the prediction of overall survival (OS) and progression-free survival (PFS), drawn from clinical, genomic, and radiomic datasets. This update directly addresses your suggestion and enhances the translational value of the review by shifting the focus from broad data categories to specific, actionable predictors of patient outcomes.

To maximise the impact, I recommend the authors create an explicit, structured table that catalogs the top 10-15 most frequently or most successfully integrated prognostic parameters (e.g., specific genomic mutations, radiomic features, or refined clinical metrics) identified by the reviewed AI/ML studies. This would transform the review into an actionable resource for subsequent clinical trial design and model development.

Response: Thank you for this valuable recommendation. In accordance with your suggestion, we have added a synthetic representation in Table 14, which summarizes the top 13 biomarkers and prognostic parameters most frequently identified across the reviewed AI/ML studies. This new table presents each feature’s primary focus, outlines its association with overall survival (OS) and progression-free survival (PFS), and briefly explains the clinical utility or methodological relevance of the models in which these parameters were applied. We believe this addition transforms the review into a more actionable and practical resource for future clinical trial design and model development in DLBCL prognostics.

Minor Comments:

IPI Comparison: The discussion adequately highlights the limitations of the IPI, but a clearer synthesis of how much AI models outperform IPI/R-IPI across various studies would strengthen the argument for adoption.

Response: Thank you for this insightful comment. We have addressed this point in the Discussion section, where we now provide a clearer synthesis of how AI/ML-based models outperform the traditional IPI and R-IPI scores across the reviewed studies. The IPI, serving as the pivotal benchmark, is now explicitly discussed in connection with the performance of other biomarkers and model-derived parameters. We highlight that the C-index values for IPI-based prediction are generally below 0.7, whereas most AI/ML models achieve superior discriminative performance, thereby reinforcing the added value of AI-driven prognostic approaches in DLBCL.

Model Transparency: A brief discussion or a section dedicated to the explainability (XAI) of the reviewed models would be beneficial, as clinical adoption requires trust and understanding of the AI's decision-making process.

Response: Thank you for this excellent and forward-looking observation. We fully agree that model transparency and interpretability are essential prerequisites for the clinical translation of AI prognostic tools. In response, we have added a new subsection in the Discussion section, entitled Model Interpretability and the Path to Clinical Translation.

This section emphasizes that the lack of explainability remains one of the main barriers to the adoption of AI in hematology and oncology, as clinical users must understand the rationale behind a model’s prediction before applying it in patient management. We highlight that the most advanced and interpretable models identified across studies include LASSO (penalized Cox regression aiding in feature selection) and XGBoost (ensemble-based methods that quantify feature importance). These approaches have helped delineate key prognostic variables such as age, ECOG performance status, and mutational profiles, as reflected in Table 14.

In contrast, we note the absence of post hoc explainability frameworks such as SHAP (SHapley Additive exPlanations) or LIME (Local Interpretable Model-agnostic Explanations) in the current literature. The revised discussion concludes by recommending that future research prioritize interpretability alongside performance, ensuring that AI-derived insights are clinically understandable, actionable, and seamlessly integrated into real-world workflows.

We sincerely thank you for your positive assessment and encouraging remarks regarding the quality and relevance of our work. We appreciate your recognition of the manuscript’s contribution to the field and are confident that the revisions addressing the synthesis of actionable prognostic parameters have further strengthened its value and alignment with the goals of precision prognostics.

Reviewer 3 Report

Comments and Suggestions for Authors

Major Comments:
          The manuscript addresses an important topic, but the presentation of the results could be improved. Currently, the manuscript describes each included study in detail, which makes it difficult for the reader to see the overall findings. A more synthesized discussion/presentation, focusing on key trends and insights rather than listing studies individually, would help readers understand the main points and their relevance.
          Graphical presentations could further improve clarity. For example, timelines of study publications could show trends in AI research over time. Other visual summaries could include distributions according to classification (e.g., modality used, patient cohort size, outcome measures), highlighting which approaches provided the most relevant clinical results.
          The manuscript needs to place greater emphasis on the outcomes of AI implementation and their practical implications for clinical practice. The authors should strengthen the synthesis of the results and include clearer visual presentations to make the findings easier to interpret and more useful for readers.

Minor Comments:
1. The order of author abbreviations (1, 3, 2) is incorrect and should be corrected.
2. The abstract is too long and should not include references.
3. The use of abbreviations throughout the manuscript is inconsistent; some abbreviations are explained multiple times, while others are not explained at all.
4. The “Materials and Methods” section should be simplified to “Methods.”
5. The “Study Selection” subsection, which is part of the Methods, currently includes results. These are repeated in the Results section and should be removed from Methods.
6. On page 5, textual data currently presented in the main text would be clearer if shown in a table.
7. In Figure 2, the text in the boxes is cut off at the bottom and should be adjusted.
8. Use of past tense is preferred throughout the manuscript. For example, on page 7, paragraph 2, the “aims” of a published study should be phrased in past tense.

Author Response

Dear Editor-in-Chief,
Dear Academic Editor and Guest Editors,
Dear Reviewer,

We greatly appreciate your careful evaluation of our manuscript entitled “Leveraging AI for Precision Prognostics in DLBCL: A Scoping Systematic Review of Emerging Approaches.” Your insightful remarks and practical suggestions have been instrumental in improving the clarity, synthesis, and overall presentation of our work. We have carefully revised the manuscript to address all comments, refining the structure, strengthening the synthesis of results, and enhancing visual clarity to better convey the key findings and their clinical significance.

Below, we provide a detailed, point-by-point response to each of your comments. All modifications made are highlighted in yellow in the revised manuscript.

Major Comments:
          The manuscript addresses an important topic, but the presentation of the results could be improved. Currently, the manuscript describes each included study in detail, which makes it difficult for the reader to see the overall findings. A more synthesised discussion/presentation, focusing on key trends and insights rather than listing studies individually, would help readers understand the main points and their relevance.
          Graphical presentations could further improve clarity. For example, timelines of study publications could show trends in AI research over time. Other visual summaries could include distributions according to classification (e.g., modality used, patient cohort size, outcome measures), highlighting which approaches provided the most relevant clinical results.
          The manuscript needs to place greater emphasis on the outcomes of AI implementation and their practical implications for clinical practice. The authors should strengthen the synthesis of the results and include clearer visual presentations to make the findings easier to interpret and more useful for readers.

Response: We sincerely thank you for these valuable observations and fully acknowledge the importance of improving presentation and enhancing reader accessibility. We completely agree that user readability and synthesis are paramount for a scoping review. However, we would like to emphasize that the primary aim of this work was to provide an exhaustive classification and consolidation of all eligible studies identified through our search strategy. The intent of this review is to systematically categorize the existing granular studies, summarize their results, performance indices, and underlying ML methods, and thereby present a comprehensive and objective overview of the current landscape.

To enhance readability and accessibility, we have designed a graphical abstract that provides a quick overview of the entire AI field in DLBCL by illustrating the distribution of studies across domains (e.g., imaging, histopathology, molecular, and clinical data). This visual component allows readers to rapidly grasp which subfields are most actively explored.

We considered including an AI research timeline; however, given the high heterogeneity and broad scope of the included studies, we found that such an addition would offer limited incremental value and could overburden the visual density of the paper. Instead, the manuscript structure has been optimized to balance narrative synthesis and tabular representation, ensuring both comprehensiveness and clarity.

Each section now integrates narrative interpretation of the studies alongside standardized tables that extract key parameters—such as input modality, model type, dataset size, and performance metrics. This dual approach enables readers to directly navigate to their area of interest, obtain a clear summary of previous findings, and compare results at a glance.

While this format is indeed more technical than a conventional narrative review, it was deliberately chosen to ensure transparency, reproducibility, and data traceability, allowing readers to engage directly with the evidence rather than with a purely interpretive synthesis. The revised version also integrates an updated synthesis of significant biomarkers (e.g., ECOG, LDH, β2-microglobulin, MYC/BCL2 alterations) that enhances the translational and clinical applicability of the findings by identifying the most recurrent predictors across AI frameworks (Table 14).

Minor Comments:
    1. The order of author abbreviations (1, 3, 2) is incorrect and should be corrected.

Response: Thank you for pointing this out. The author abbreviations have been rechecked and updated to follow the correct numerical order and to comply with MDPI publication standards.

  1. The abstract is too long and should not include references.

Response: Thank you for your observation. All references have been removed from the abstract, and its length has been optimized to ensure conciseness and compliance with the journal’s formatting requirements.

  1. The use of abbreviations throughout the manuscript is inconsistent; some abbreviations are explained multiple times, while others are not explained at all.

Response: Thank you for this helpful comment. All abbreviations have been thoroughly reviewed and standardised throughout the manuscript. Their use has been optimised in both the abstract and the main text, ensuring that each abbreviation is defined only once upon first appearance and applied consistently thereafter.

  1. The “Materials and Methods” section should be simplified to “Methods.”

Response: Thank you for this suggestion. We have complied accordingly, and the section title has been simplified from “Materials and Methods” to “Methods.”

  1. The “Study Selection” subsection, which is part of the Methods, currently includes results. These are repeated in the Results section and should be removed from Methods.

Response: Thank you for this valid observation. We have addressed this point by removing the result-related content from the Study Selection subsection in the Methods section. We appreciate this insightful comment, which has helped improve the overall structure and narrative flow of the manuscript.

  1. On page 5, the textual data currently presented in the main text would be clearer if shown in a table.

Response: We appreciate this thoughtful suggestion. However, we believe that converting the specified textual content into an additional table would further increase the number of tables in a manuscript that is already data-dense. To maintain visual balance and readability, we have opted to preserve this information in its current narrative form, which we consider sufficiently clear within the overall structure of the paper.

  1. In Figure 2, the text in the boxes is cut off at the bottom and should be adjusted.

Response: Thank you for this observation. Figure 2, representing the PRISMA flow chart, will be provided during the editorial process as a standalone high-resolution image to ensure full visibility and correct formatting of all text elements within the boxes.

  1. Use of past tense is preferred throughout the manuscript. For example, on page 7, paragraph 2, the “aims” of a published study should be phrased in past tense.

Response: Thank you for this helpful comment. The manuscript has been carefully reviewed multiple times to ensure consistent use of past tense when referring to previously published studies, as well as to correct any remaining grammatical issues, typographical errors, or stylistic inconsistencies.

We sincerely thank you for your positive assessment and encouraging remarks regarding the quality and relevance of our work.

Round 2

Reviewer 1 Report

Comments and Suggestions for Authors

The article can be published.

Author Response

Thank you for your dedicated time and valuable suggestions.

Reviewer 2 Report

Comments and Suggestions for Authors

I accept as is.

Author Response

(The authors gave the same response as above.)

Reviewer 3 Report

Comments and Suggestions for Authors

Thank you to the authors for their response and for revising the manuscript.

According to the authors’ clarification, the primary aim of this work was to provide an exhaustive classification and consolidation of all eligible studies identified through their search strategy, as the intent of the review is to systematically categorise existing granular studies, summarise their results, performance indices, and underlying machine learning methods, and thereby present a comprehensive and objective overview of the current landscape. Given this stated aim, I suggest that the title be adjusted accordingly. For instance, a title such as “Systematic Categorisation and Performance Analysis of Artificial Intelligence and Machine Learning Models in Studies of Diffuse Large B-Cell Lymphoma: A Scoping Review” (or a similar formulation) would more accurately reflect the study’s scope and objectives.

The Discussion section includes is now extended, but additional subsections (ex 4.2, 4.3) and a table (Table 14) that appear to fit more appropriately within the Results section of the scoping review. The authors may ensure that this section follows the conventional structure and focus expected for a discussion and distribute the scientific information more accurately. Conclusion shall be a separate chapter.

Also, authors shall consider not using strong statements within the manuscript. While the use of tools such as ChatGPT is acknowledged, the authors should maintain scientific simplicity in data presentation and phrasing (general observation).

Minor comments

Several paragraphs lack supporting references (e.g., lines 424–431 and 935–952). Please ensure that all statements, particularly those summarising or interpreting previous studies, are adequately referenced.

At line 432, the authors should clearly state that only one study was identified and then provide a concise description of its key characteristics and findings.

Lines 997–999: The authors state, “As a scoping review, with a heterogeneous database of studies, we decided, in line with the PRISMA guidance, that a formal risk of bias assessment would not have provided additional valuable insights.” Authors cannot base a decision on something not investigated. Use rather assumed, speculated…

Figure 1: Since the figure is already described above, there is no need for a title within the figure itself. If the figure was generated using AI, this must be explicitly stated in the figure legend.

Table 5: The phrase “this is” within the first row of the table should not be used if other tables employ the passive voice. Please correct

At line 1009, the phrase “As a final thought” is not appropriate for scientific writing, neither a starting tone of a Conclusion section.

Author Response

We thank the Academic Editor and the reviewers for the detailed and constructive process. We have carefully considered each point of feedback, and we believe the resulting manuscript revisions effectively address all concerns and strengthen the overall clarity and scientific contribution of our systematic review.

Comment 1 : According to the authors’ clarification, the primary aim of this work was to provide an exhaustive classification and consolidation of all eligible studies identified through their search strategy, as the intent of the review is to systematically categorise existing granular studies, summarise their results, performance indices, and underlying machine learning methods, and thereby present a comprehensive and objective overview of the current landscape. Given this stated aim, I suggest that the title be adjusted accordingly. For instance, a title such as “Systematic Categorisation and Performance Analysis of Artificial Intelligence and Machine Learning Models in Studies of Diffuse Large B-Cell Lymphoma: A Scoping Review” (or a similar formulation) would more accurately reflect the study’s scope and objectives.

Response: We thank the reviewer for this constructive and insightful suggestion. We fully agree that the original title did not adequately reflect the core objective of the work, which is systematic classification and performance reporting. The proposed rationale perfectly captures the scope of our scoping review. Therefore, we have modified the manuscript title to accurately reflect its core objective of categorizing models and analyzing their performance metrics. The new title is: “Artificial Intelligence for Risk Stratification in Diffuse Large B-Cell Lymphoma: A Systematic Review of Classification Models and Predictive Performances”. We believe this title is more comprehensive and directly integrates the key terms and methodology, aligning with the reviewer’s suggestion and the stated aims of the review.

Comment 2 : The Discussion section includes is now extended, but additional subsections (ex 4.2, 4.3) and a table (Table 14) that appear to fit more appropriately within the Results section of the scoping review. The authors may ensure that this section follows the conventional structure and focus expected for a discussion and distribute the scientific information more accurately. Conclusion shall be a separate chapter.

Response: We thank the reviewer for pointing out the need for a more conventional and rigorous structure, particularly for a scoping review following PRISMA-ScR guidelines. We agree that the detailed, descriptive synthesis of the included studies is more accurately placed within the Results section. The manuscript has been restructured in line with the reviewer's suggestion. Specifically: - Content Relocation: The descriptive subsections containing the systematic categorization and synthesis of the included studies (including the data from the previous Table 14) have been transitioned from the former Discussion section into the Results chapter. This ensures that the presentation of the scientific findings (the "mapping" of the literature) precedes its interpretation. - New Conclusion Section: A distinct, new chapter titled Conclusions has been added at the end of the manuscript, providing a final, separate summary of the review's main findings and implications. This change aligns the document with the standard academic structure for a systematic review.

Comment 3 : Also, authors shall consider not using strong statements within the manuscript. While the use of tools such as ChatGPT is acknowledged, the authors should maintain scientific simplicity in data presentation and phrasing (general observation).

Response: We acknowledge and appreciate this general but crucial observation. We fully agree on the importance of maintaining scientific objectivity, simplicity, and rigor in all academic phrasing and data presentation. We intend to ensure the tone remains professional and measured throughout the manuscript. We have performed a comprehensive review of the entire manuscript's language and style. All instances of potentially "strong" or non-academic phrasing have been moderated to align with the objective, formal tone expected in a systematic review. We confirm that a final, meticulous check for language and style has been performed across all sections to ensure scientific simplicity and accuracy in data presentation, thus adhering to the standards of academic reporting.

Minor comments

Comment 4 : Several paragraphs lack supporting references (e.g., lines 424–431 and 935–952). Please ensure that all statements, particularly those summarising or interpreting previous studies, are adequately referenced.

Response: We fully agree that rigorous and accurate referencing is essential, especially for a systematic review where the narrative synthesis must be clearly attributed to the source studies. We have thoroughly reviewed the manuscript to ensure all statements are correctly supported by the cited literature. The specific paragraphs mentioned (e.g., those around lines 424–431 and 935–952, among others) have been checked, and the appropriate references have been assigned. This correction ensures that all summaries, interpretations, and discussions of previous studies are accurately and robustly referenced, maintaining the high standard of academic integrity.

Comment 5 : At line 432, the authors should clearly state that only one study was identified and then provide a concise description of its key characteristics and findings.

Response: We thank the reviewer for highlighting the need for this clarification. We agree that explicitly stating the presence of a singular study enhances the precision and objectivity of the data modality section. Furthermore, the subsequent paragraph provides a concise, detailed description of that singular study's key characteristics, methodology (automated CACS scoring + regression models), and main findings (OR for CTRCD and SHR for MACE).

Comment 6 : Lines 997–999: The authors state, “As a scoping review, with a heterogeneous database of studies, we decided, in line with the PRISMA guidance, that a formal risk of bias assessment would not have provided additional valuable insights.” Authors cannot base a decision on something not investigated. Use rather assumed, speculated…

Response: We thank the reviewer for this critical observation regarding the language used to justify the exclusion of a formal risk of bias assessment. We agree that the original phrasing implied a conclusion based on insufficient investigation and lacked the necessary scientific precision. The text has been revised to remove the strong, problematic statement. The updated paragraph now uses more precise and objective terminology to describe the approach, clarifying that, consistent with the methodology of a scoping review and PRISMA-ScR guidance, a descriptive mapping of the literature was prioritized over a formal, quantitative risk of bias assessment. This change ensures the methodology is presented with rigorous academic language and avoids problematic phrasing.

Comment 7 : Figure 1: Since the figure is already described above, there is no need for a title within the figure itself. If the figure was generated using AI, this must be explicitly stated in the figure legend.

Response: We thank the reviewer for this formatting and style guidance. We agree that the figure should conform to the specific standards of scientific publishing (e.g., MDPI guidelines) regarding titles and disclosure of generative AI use. The figure's formatting has been updated as requested. The standalone title within the figure area has been removed. Furthermore, the figure legend (Figure 1) has been explicitly updated to state that the graphic was generated using an AI tool (napkin.ai), ensuring full transparency and adherence to journal requirements.

Comment 8 : Table 5: The phrase “this is” within the first row of the table should not be used if other tables employ the passive voice. Please correct

Response: We appreciate the reviewer's attention to detail regarding consistency in table phrasing. We agree that maintaining a uniform and objective voice across all tables is essential for scientific clarity. This correction has been implemented. The phrasing in the first row of Table 5 has been revised to remove the phrase “this is,” ensuring the language is consistent with the style (passive or direct objective phrasing) used in other summary tables throughout the manuscript.

Comment 9 : At line 1009, the phrase “As a final thought” is not appropriate for scientific writing, neither a starting tone of a Conclusion section.

Response: We thank the reviewer for this precise and valuable editorial guidance. We fully agree that the phrase is not appropriate for the formal tone of scientific writing, especially when introducing the Conclusions section. The text has been replaced with a more formal and objective opening statement that adheres to the required standard of academic writing.

We sincerely thank the Academic Editor and the reviewers once more for their time, rigorous evaluation, and constructive feedback on our manuscript. We believe the revisions have significantly improved the clarity and scientific rigor of our work.

Round 3

Reviewer 3 Report

Comments and Suggestions for Authors

Thank you for addressing my comments, I have no further suggestions.

I have only one small observation: I suggest you change the section title “4. Discussions, Future Perspectives:” to “4. Discussion”. The subsections beneath are ok as they are.